# Targeting natural splicing plasticity of APOBEC3B restricts its expression and mutagenic activity

A. Rouf Banday[1], Olusegun O. Onabajo [1], Seraph Han-Yin Lin[1], Adeola Obajemu[1], Joselin M. Vargas [1], Krista A. Delviks-Frankenberry [2], Philippe Lamy [3], Ariunaa Bayanjargal[1], Clara Zettelmeyer[1], Oscar Florez-Vargas [1], Vinay K. Pathak [2], Lars Dyrskjøt [3] & Ludmila Prokunina-Olsson [1✉]

APOBEC3A (A3A) and APOBEC3B (A3B) enzymes drive APOBEC-mediated mutagenesis. Identification of factors affecting the activity of these enzymes could help modulate mutagenesis and associated clinical outcomes. Here, we show that canonical and alternatively spliced A3A and A3B isoforms produce corresponding mutagenic and non-mutagenic enzymes. Increased expression of the mutagenic A3B isoform predicted shorter progression-free survival in bladder cancer. We demonstrate that the production of mutagenic vs. non-mutagenic A3B protein isoforms was considerably affected by inclusion/skipping of exon 5 in A3B. Furthermore, exon 5 skipping, resulting in lower levels of mutagenic A3B enzyme, could be increased in vitro. Specifically, we showed the effects of treatment with an SF3B1 inhibitor affecting spliceosome interaction with a branch point site in intron 4, or with splice-switching oligonucleotides targeting exon 5 of A3B. Our results underscore the clinical role of A3B and implicate alternative splicing of A3B as a mechanism that could be targeted to restrict APOBEC-mediated mutagenesis.

[1] Laboratory of Translational Genomics, Division of Cancer Epidemiology and Genetics, National Cancer Institute, National Institutes of Health, Bethesda, MD, USA. [2] Viral Mutation Section, HIV Dynamics and Replication Program, Center for Cancer Research, National Cancer Institute, Frederick, MD, USA. [3] Department of Molecular Medicine, Aarhus University Hospital, Aarhus, Denmark. ✉email: prokuninal@mail.nih.gov

Enrichment of C-to-T or C-to-G mutations within the TCA and TCT motifs, attributed to APOBEC-mediated mutagenesis, has been implicated in cancer susceptibility[1], tumor evolution[2,3], metastatic progression[4,5], treatment response[3,6], and survival[1]. Thus, understanding and harnessing the mechanisms regulating this mutational process is of clinical importance. Among the seven members of the APOBEC3 enzyme family, A3A[7,8], A3B[9], and an allelic variant of APOBEC3H (A3H)[10] have been linked with APOBEC mutagenesis in tumors.

The intrinsic and extrinsic factors that regulate the expression levels of APOBEC3s might explain some of the differences in a load of APOBEC-signature mutations within and between tumors of different types. These intrinsic factors include common germline variants—a single-nucleotide polymorphism (SNP) rs1014971 and the polymorphic A3AB deletion[1,11], or their correlated proxies, SNPs rs17000526 and rs12628403[1]. The same genetic variants have also been associated with A3B expression. Specifically, rs1014971 regulates A3B expression through an allele-specific effect on an enhancer upstream of the APOBEC3 gene cluster[1] and the A3AB deletion—through the elimination of one or both copies of the A3B gene[11,12]. A haplotype represented by a missense SNP rs139297 (Gly105Arg) that creates an A3H protein isoform with nuclear localization (A3H-I) has been associated with APOBEC-signature mutations in A3A and A3B-null breast and lung tumors[10]. Extrinsic factors that induce expression of specific APOBEC3s include viral infections[1,13] and exposure to environmental or chemotherapeutic DNA-damaging agents[1,2,14]. Most conclusions about the role of A3A and A3B in mutagenesis were drawn based on RNA-seq studies despite the poor ability to resolve and confidently quantify mRNA expression of these highly homologous APOBECs by short sequencing reads[10].

Alternative splicing (AS) of pre-mRNA can produce functionally distinct isoforms or regulate gene expression through downstream mechanisms, such as nonsense-mediated decay (NMD)[15]. Alternatively spliced isoforms of other APOBEC3 genes (APOBEC3H and APOBEC3F) have been reported to generate enzymes with variable activity[16–18], but the effects of AS in A3A and A3B on functional activities of these enzymes and relevant clinical outcomes have not been explored. Here, we characterized AS in A3A and A3B in relation to APOBEC-mediated mutagenesis and explored the mechanisms of its regulation and possible therapeutic modulation.

## Results

### Expression profiling of A3A and A3B transcripts: quantification of exon–exon junctions vs. total gene expression.

According to the human reference genome annotation (hg19, UCSC), A3A and A3B genes have two and three alternative isoforms, respectively (Supplementary Table 1 and Supplementary Fig. 1a). Among these, only canonical isoforms, which we designate as A3A1 and A3B1, but not alternatively spliced isoforms (A3A2, A3B2, and A3B3), have been studied. First, we confirmed the existence of all these isoforms by analyzing RNA-seq reads in the TCGA dataset (see Supplementary Table 2 for samples used in the analysis) and identifying exon–exon junction reads specific to each of these isoforms (Fig. 1a and Supplementary Fig. 1b). The alternative A3A and A3B isoforms were expressed in 3.49% and ~50% samples with mean read counts of 15.4 and 5.2, respectively (Fig. 1b, c). In Supplementary Note 1, we show that due to the high homology between A3A and A3B, quantification based on specific exon–exon junctions (by RNA-seq or qRT–PCR) is more reliable than based on total gene expression analysis (by RNA-seq). The misaligned RNA-seq reads would incorrectly represent the expression of both genes, affecting downstream analyses and biological interpretations.

Specifically, based on total RNA-seq reads, A3B was undetectable only in 0.16% (18 of 11,058) of TCGA samples (Fig. 1b). However, based on exon–exon junction reads for the canonical isoform A3A1 (E1–E2 junction), expression was undetectable in 36.9% (4079 of 11,058) of TCGA samples (Fig. 1b). Similarly, A3B expression was undetectable by total RNA-seq reads only in 0.07% (8 of 11,058) of TCGA samples, but this number was 4.2% (468 of 11,058) based on exon–exon junction reads for the canonical A3B1 isoform (E5–E6 junction reads, Fig. 1c).

**Alternative protein isoforms of A3A and A3B are non-mutagenic.** To better understand the functional properties of these alternative isoforms, we performed computational analysis of their corresponding protein sequences. Compared to A3A1, A3A2 lacks 18 aa (residues 10–28) due to AS between exons 1 and 2, including residues His11 and His16, which are important for deamination activity[19] (Fig. 2a). A3B2 is produced due to AS in exon 6, resulting in the loss of 25 aa (residues 242–266, Fig. 2b), including His253 that stabilizes the zinc cofactor, and Glu255 that directly participates in a nucleophilic attack on cytosine during deamination[20].

The A3B3 isoform is generated by skipping of exon 5 in A3B and encodes truncated and likely unstable protein without the catalytic domain, in which a 132 aa fragment in the C terminus of A3B is replaced by 62 aa of an aberrant frame-shifted sequence (Fig. 2b). The stop codon in the penultimate exon also makes the A3B3 transcript a potential target for NMD. Indeed, A3B3 mRNA expression was significantly increased after treatment of HT-1376 cells with digoxin, a known NMD inhibitor[21] (Fig. 2c). Due to NMD, expression studies, including by RNA-seq, would detect only residual levels of A3B3, underestimating its expression.

Evaluation of the deaminating (mutagenic) potential of recombinant APOBEC3 proteins (Fig. 2d) showed activities for the canonical A3A1 and A3B1 isoforms, but not for the alternative A3A2, A3B2, and A3B3 proteins (Fig. 2e, f). The alternative protein isoforms did not show dominant-negative effects on the activities of A3A1 and A3B1 evaluated either by deamination assays (Supplementary Fig. 2) or by HIV-1 infectivity inhibition assays[22–24] (Fig. 2g and Supplementary Fig. 3). Based on these results, we conclude that AS of A3A and A3B results in the production of catalytically inactive, non-mutagenic protein isoforms of these enzymes.

**Isoform-specific analysis refines the correlations between A3A and A3B expression and APOBEC-mediated mutagenesis.** Previously, multiple reports showed correlations between the total expression of A3A and A3B genes and the burden of APOBEC-signature mutations in several tumor types[8,25]. Now, we revisited these conclusions based on correlations between the expression of the mutagenic A3A1 and A3B1 isoforms quantified by RNA-seq read counts for specific exon–exon junctions and "APOBEC mutation pattern" proposed as the most stringent estimate of APOBEC-signature mutation burden[7]. The analysis was performed in six cancer types, with ≥10% of tumors carrying these mutations (Supplementary Fig. 4).

We observed significant positive correlations between the expression of both mutagenic isoforms—A3A1 and A3B1 and APOBEC mutation burden in several cancer types. A3A1-specific correlations were observed in BRCA ($P = 5.49E-08$; $rho = 0.17$), CESC ($P = 7.84E-08$; $rho = 0.37$), and HNSC ($P = 3.684E-08$; $rho = 0.24$), while A3B1-specific correlations in BLCA ($P = 2.76E-07$; $rho = 0.25$). These results corroborate previous findings based on germline variants, which implicated A3A as a more prominent mutagen in BRCA and A3B—in BLCA[1,11].

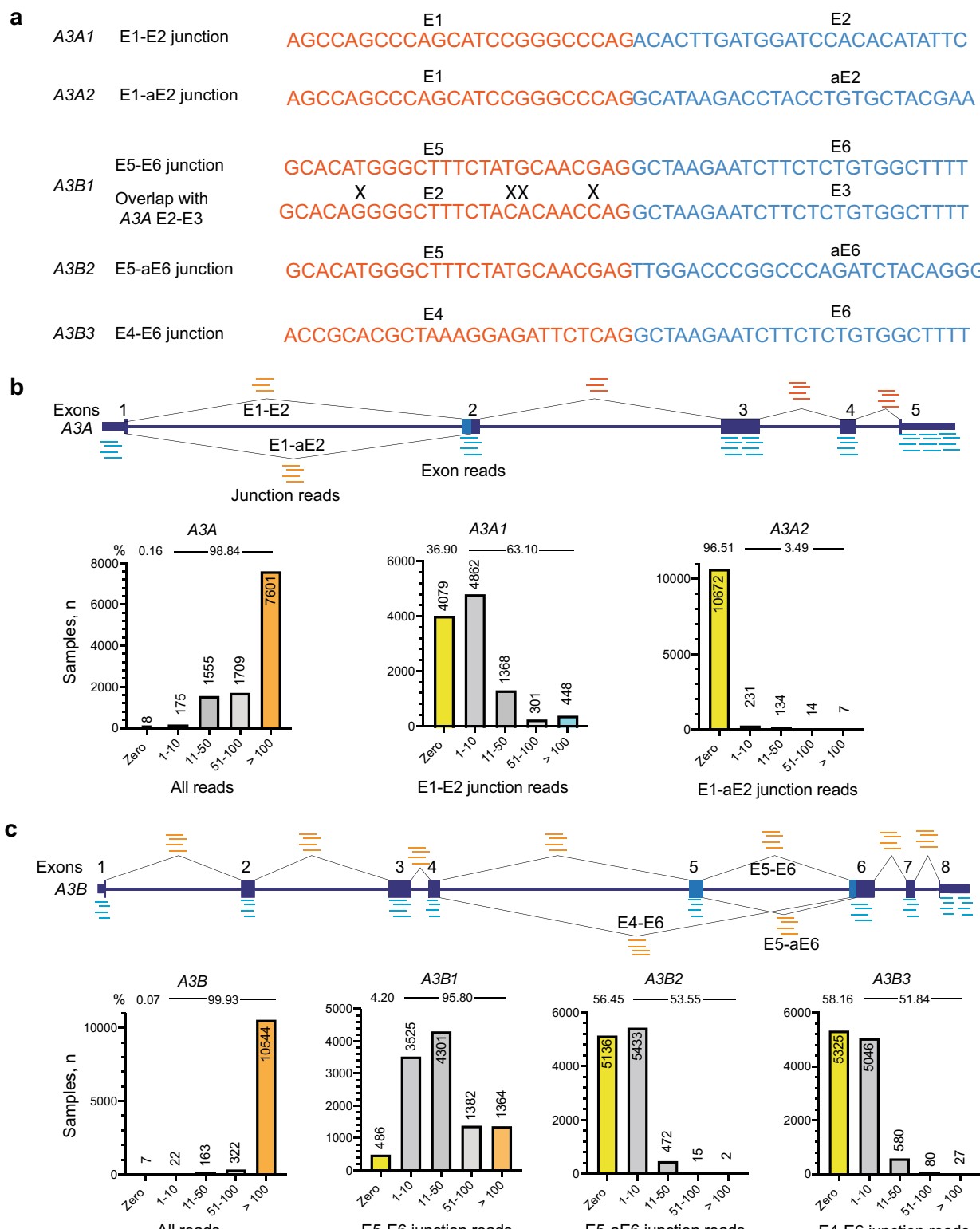

**Fig. 1 Quantification of *A3A* and *A3B* expression based on the total and exon–exon junction RNA-seq reads in 11,058 TCGA samples. a** Nucleotide sequences of exon–exon junctions specific to *A3A* and *A3B* isoforms. "X" – mismatch between *A3A* and *A3B* sequences. **b** Schematics of *A3A* exons and splicing junctions. The bar graphs show the numbers of samples (*y* axis) in relation to RNA-seq read counts grouped in 5 sub-categories (*x* axis) for the total *A3A* expression and exon–exon junction-based expression of *A3A1* and *A3A2* isoforms. 'E' refers to exon and 'aE' refers to an alternative exon. Based on exon junction reads (≥1 read/sample), *A3A1* (E1–E2 junction) and *A3A2* (E1-aE2 junction) is expressed in 6979 (63.10%) and 386 (3.49%) TCGA samples, respectively. **c** Schematics of *A3B* exons and splicing junctions and comparison of gene expression based on the total and exon–exon junction reads corresponding to *A3B* isoforms. *A3B1* is expressed in 10,572 (95.8%) TCGA samples, but A3B2 and A3B3 are expressed in 53.55% (5922 of 11,058) and 51.84% (5733 of 11,058) of TCGA samples, respectively. The TCGA RNA-seq set includes 10,328 tumors and 730 adjacent normal tissue samples.

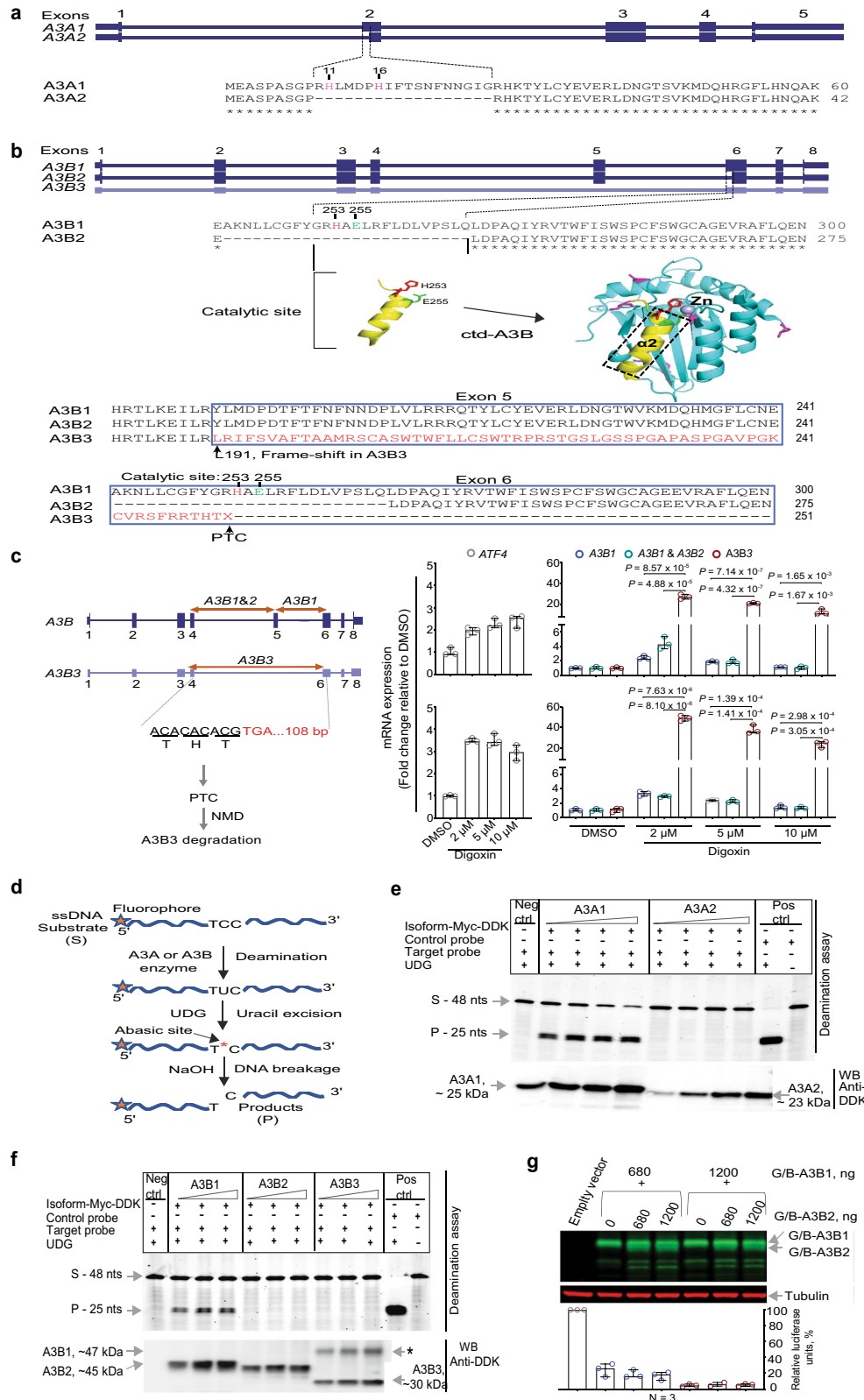

In LUAD, the correlations were comparable for both *A3A1* (P = 3.12E−12; rho = 0.31) and *A3B1* (P = 7.73E−12; rho = 0.30) and in LUSC—the correlation was moderately significant only for *A3A1* (P = 1.86E−02; rho = 0.17) (Fig. 3a, b). Notably, these tumor-specific correlations were not reported by studies based on total gene expression[8,25].

**A3B as a driver of APOBEC-mediated mutagenesis and a predictor of progression-free survival in bladder cancer.** It was reported that episodes of APOBEC-mediated mutagenesis could be observed in cell lines propagated in culture[26]. Those included bladder cancer cell lines, in which we detected the expression of *A3B1* but not *A3A1* (Supplementary Note 1), thus nominating

**Fig. 2 Alternative splicing in *A3A* and *A3B* results in catalytically inactive protein isoforms.** Clustal Omega alignment of protein sequences for **a** canonical A3A1 and alternative A3A2 and **b** canonical A3B1 and alternative A3B2 and A3B3 isoforms. A3A2 lacks 18 N-terminal aa (R10 - G27), including functional residues H11 and H16[19], and A3B2 lacks a fragment with functional residues E255 and H253[20]. Skipping of exon 5 generates an *A3B3* transcript that encodes a truncated protein without the catalytic domain, in which a frameshift at position L191 results in the replacement of a 132 aa fragment in the C terminus of A3B by 62 aa of an aberrant frame-shifted sequence. **c** Due to a premature termination codon (PTC) in the penultimate exon 6, *A3B3* might be targeted by nonsense-mediated decay (NMD). The effect of NMD demonstrated by analysis of mRNA expression of *ATF4* (positive control) and *A3B* isoforms with TaqMan expression assays indicated by arrows in HT-1376 cells treated with DMSO (vehicle) or digoxin, an NMD inhibitor. **d** Outline of the in vitro deamination assays testing conversion of a 48-nucleotide (nt) ssDNA substrate (S) into a 25-nt product (P) by the recombinant C-terminally Myc-DDK tagged protein isoforms **e** A3A and **f** A3B. Negative control reactions (Neg ctrl) lack A3A and A3B proteins; positive control probe (Pos ctrl) is completely converted by the UDG enzyme. Western blot (WB) analysis with an anti-DDK antibody shows the amounts of A3A and A3B proteins in reactions. **f** Asterisk indicates a non-specific band resulting from deamination of ssDNA probe by A3B3. Because DNA binding domain is intact in A3B3, the higher molecular weight (MW) band could indicate a probe bound by A3B3. **g** HIV-1 infectivity restriction assays show no inhibitory effects of A3B2 on the activity of the A3B1 protein isoform measured for A3G/A3B protein fusions (G/B). Shown one of three representative experiments with Western blot for corresponding recombinant proteins and normalized relative luciferase units (%) for each labeled condition. All bar graphs (**c**, **g**) show individual data points, with error bars representing 95% confidence interval for the medians, based on three biological replicates. *P*-values are for the two-sided unpaired Student's *t*-test.

A3B1 rather than A3A1 as the primary mutagenic APOBEC3 enzyme responsible for these episodes of mutagenic activity.

Our analysis of non-muscle-invasive bladder tumors from the UROMOL study[27] further supported the role of A3B1 in bladder cancer. Specifically, using the same RNA-seq exon junction isoform-specific quantification of expression as in TCGA samples, we observed that higher expression of *A3B1* was significantly associated with increased APOBEC-mediated mutation burden ($P = 5.51E-06$) and with shorter progression-free survival ($P = 5.69E-05$) in patients with non-muscle-invasive bladder cancer (Fig. 3c, d). Similar quantification of the *A3A1* expression in the UROMOL tumors showed no association with APOBEC-mediated mutagenesis or progression-free survival (Fig. 3d).

**Non-mutagenic isoform A3B3 is more common in normal tissues compared to tumors.** Alternative *A3B* isoforms (*A3B2* and *A3B3*) were detected in ≥50% of 11,058 TCGA samples (Fig. 1), albeit at very low levels compared to canonical *A3B1* isoform. Although these isoforms may not produce any functional proteins (Fig. 1), they could still be important. For example, splicing towards alternative, non-mutagenic isoforms might decrease the production of the canonical, mutagenic isoforms. Because mutagenesis is considered tumorigenic, we hypothesized that the proportion of splicing towards non-mutagenic isoforms might be higher in normal tissues compared to tumors. To test this hypothesis, we calculated the percent spliced-in index (PSI)[28] based on RNA-seq read counts of specific exon–exon junctions in TCGA paired tumor and adjacent normal tissues. *A3A* was excluded from this analysis because the expression of the alternative *A3A2* isoform was detected only in 2.57% (18 of 698) of normal samples (Supplementary Fig. 5a). Thus, we calculated PSI for exons 5 and 6 of *A3B* in 17 TCGA cancer types with ≥5 tumor/normal pairs. We observed that the proportion of *A3B* canonical splicing was higher, while the proportion of alternative splicing was lower in tumors compared to corresponding adjacent normal tissues.

Specifically, the proportion of *A3B2* splicing was lower in tumors of KICH ($P = 1.0E-03$) and LUSC ($P = 3.30E-02$) (Supplementary Fig. 5b) and the proportion of *A3B3* splicing was lower in tumors of BLCA ($P = 6.65E-03$), HNSC ($P = 7.09E-03$), LIHC ($P = 6.01E-03$), LUAD ($P = 6.69E-04$), and LUSC ($P = 5.55E-07$) (Supplementary Fig. 5c). All other cancer types showed no significant differences in this analysis (Supplementary Fig. 5d, e). Most cancer types with a low proportion of non-mutagenic isoforms, specifically of *A3B3*, also had a higher rate of APOBEC-mediated mutation burden[7,8], with exceptions

for tumors of KICH and LIHC, in which loads of APOBEC-signature mutations were negligible (Supplementary Fig. 4). In these cancers, A3B might play a mitogenic rather than a mutagenic role as has been shown for hepatocellular carcinoma[29] and suggested for breast cancer[30].

We also performed RT-PCR analysis and sequencing of splicing junctions between exons 4 and 6 of *A3B* in a panel of 33 muscle-invasive bladder tumors and 30 adjacent normal tissues (Fig. 4a). Exon 5 skipping, generating *A3B3*, was the most frequently observed AS event, similar to the pattern observed in TCGA, present in 30% of normal tissues compared to 13% of tumors (Fig. 4a). Western blot analysis in the same tissue samples showed that A3B1 protein expression was common in tumors but rare in normal tissues, although the frame-shifted and truncated A3B3 protein could not be detected (Fig. 4b and Supplementary Note 2). We then used isoform-specific TaqMan assays to quantify the expression of A3B isoforms in the same set of bladder tissues. *A3B1* was expressed significantly higher in tumors, while *A3B3* was higher in normal tissues, and *A3B2* was not quantifiable (Fig. 4c). The fact that AS of *A3B* was more common in normal tissues suggests that increased generation of alternative, non-mutagenic A3B isoforms might be anti-tumorigenic or/and inhibited in tumors.

**A3B exon 5 skipping is sensitive to expression levels of splicing factors.** Considering that A3B1 is clinically relevant, at least in bladder cancer (Fig. 3c, d), decreasing its expression might be of therapeutic importance. We hypothesized this could be achieved by shifting *A3B* pre-mRNA splicing from the mutagenic *A3B1* to non-mutagenic *A3B2* or *A3B3* isoforms. To this end, we first sought to explore the regulation of these splicing events. AS outcomes are regulated by the interaction of *trans*-acting spliceosomal and splicing factors (SFs) with *cis*-acting intronic and exonic pre-mRNA motifs[31]. To explore AS of *A3B*, we created mini-genes by cloning the corresponding alternative exons with 80 bp of flanking intronic sequences into an Exontrap vector (Supplementary Fig. 6a). These mini-genes were transiently transfected into HEK293T cells, and their splicing patterns were evaluated.

The mini-gene experimental system was not informative for evaluation of splicing occurring via a cryptic splicing site in exon 6 of *A3B*, as the observed splicing pattern represented only the canonical (*A3B1*) but not the alternative (*A3B2*) isoform (Supplementary Fig. 6b). However, we could capture the AS pattern of *A3B* exon 5, with its inclusion representing a proxy of the *A3B1* isoform and its skipping representing a proxy of the *A3B3* isoform. We used this mini-gene for exon 5 (E5) to further explore the regulation of *A3B1* versus *A3B3* splicing patterns.

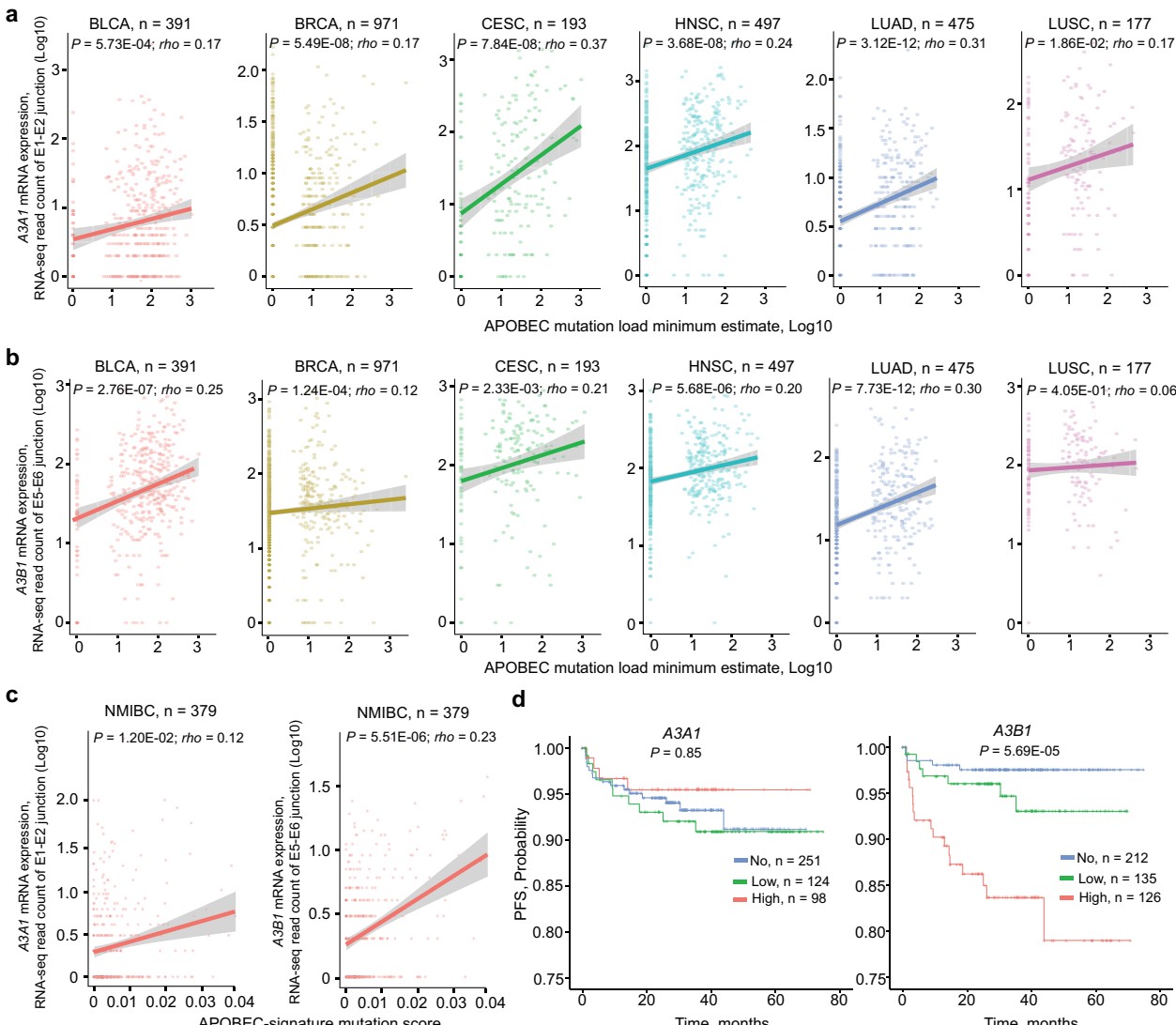

**Fig. 3 Analysis of *A3A1* and *A3B1* expression based on exon–exon junction RNA-seq reads in relation to APOBEC-mediated mutagenesis and progression-free survival in patients with non-muscle-invasive bladder cancer (NMIBC).** Correlation analysis between mRNA expression of **a** *A3A1* and **b** *A3B1* and APOBEC-mediated mutagenesis, measured as "APOBEC mutation load minimum estimate" in six cancer types in TCGA. Spearman correlation coefficients with *P*-values for two-sided tests are shown. The line in scatterplots represents the linear trend and the shading represents the 95% confidence interval around the line of best fit. **c** Significant correlation between APOBEC-signature mutation score (S3) and *A3B1* but not *A3A1* expression in non-muscle-invasive bladder cancer (NMIBC) in the UROMOL study. **d** Kaplan–Meier plots for progression-free survival (PFS) of NMIBC based on *A3A1* and *A3B1* isoform expression in the UROMOL study. *P*-values are for multivariable Cox-regression models adjusted for sex, age, and tumor stage. Grouping into "No", "Low", and "High" groups was done based on *A3B1* RNA-seq read counts, separating samples with no expression (zero reads) and then below and above the median for the remaining samples.

We hypothesized that splicing of *A3B* exon 5 might be affected by SFs and bioinformatically predicted several candidate SFs that bind within this exon (Supplementary Table 3). To experimentally test these predictions, we co-transfected HEK293T cells with expression constructs for 10 of these SFs (Supplementary Table 4 and Fig. 5a) together with the E5 mini-gene. Of these, four SFs—SRSF2, SRSF3, CELF1-T4, and ELAVL2—significantly affected exon 5 splicing (Fig. 5b), with SRSF2 showing the strongest effect. The effect of SRSF2 on exon 5 splicing was further confirmed in three additional bladder cancer cell lines—HT-1376, SW-780, and HTB-9 (Supplementary Fig. 7). Screening of exon 5 splicing in 10 cell lines of different tissue origin also showed variable exon 5 skipping (Fig. 5c), presumably due to differences in levels of expression/activity of endogenous SFs in these cell lines.

To test this, we performed a correlation analysis between the percentage of *A3B* exon 5 skipping in the mini-gene experiment (Fig. 5c) and total mRNA expression of the eight SFs evaluated in this experiment (Fig. 5c), using RNA-seq data available in the Cancer Cell Line Encyclopedia (CCLE). HEK293T cell line was excluded from this analysis because RNA-seq data for this cell line was not available in CCLE. *A3B* exon 5 skipping showed significant positive correlations with the expression of *SRSF2*, *KHDRBS1,* and *MBNL1*, and negative correlations with the expression of *ELAVL2* (Supplementary Note 5). The correlation with *SFRS2* expression was the strongest and supported the results of the overexpression model (Fig. 5b). Thus, our results suggested that *A3B* exon 5 skipping is sensitive to expression levels of multiple SFs, which can bind to cis-regulatory motifs within this exon.

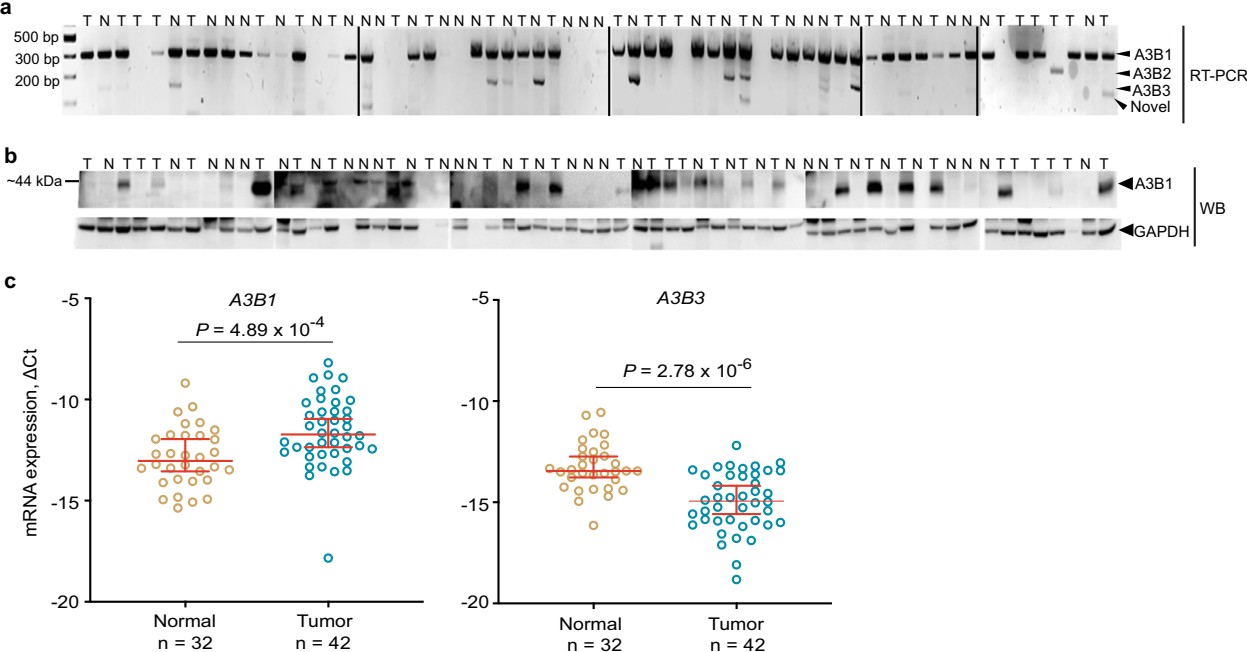

**Fig. 4 Analysis of *A3B* alternative splicing in paired tumor and adjacent normal samples. a** RT-PCR analysis in 63 bladder tissue samples. Sanger sequencing of RT-PCR products generated with primers for *A3B* exons 4 and 6 shows the canonical (*A3B1*) and alternative (*A3B2* and *A3B3*) splicing events. A splicing event (labeled as "Novel") that involves both exon 5 skipping and the use of a cryptic splice site in exon 6, was detected in three samples. An AS event (mainly *A3B3*) was observed in 19% of all samples (13 of 63 samples), including 13% of tumors (4 of 30) and 27% of adjacent normal tissues (9 of 33). **b** Western blot analysis for the A3B1 protein. **c** qRT–PCR analysis for *A3B1* and *A3B3* isoforms. mRNA expression levels of *A3B1* were significantly higher in tumors and of *A3B3*—in adjacent normal tissues. *A3B2* expression was not detected in >95% samples (Supplementary Data 1). Plots show all individual data points with error bars representing 95% confidence intervals for the medians. *P*-values are for the non-parametric Mann–Whitney *U* test.

**Skipping of *A3B* exon 5 is facilitated by weak intronic branch point sites within intron 4.** Bioinformatics analysis of *A3B* showed differences in the distribution of predicted cis-regulatory motifs - intronic branch point sites (BPS) (Fig. 5d and Supplementary Table 5) and exonic splicing silencers/enhancers (Fig. 5e). The most striking differences were found within intron 4 that harbored the fewest and the weakest BPS of those scored in all *A3B* introns. The highest scored intronic BPS located 38 and 50 bp upstream of exon 5 were included in the 80 bp of flanking intronic sequences in the E5 mini-gene (referred to as E5$^{BPS+}$) and supported the observed partial inclusion of exon 5 that generates the *A3B1*-type canonical splicing event (Fig. 5e).

We then created an E5 mini-gene version with only 20 bp of flanking intronic sequences to exclude the putative BPS and observed no canonical *A3B1*-type splicing in this model (E5$^{BPS−}$, Fig. 5e), confirming the importance of the putative BPS in intron 4 for exon 5 inclusion/skipping. In agreement with the stronger predicted BPS in intron 5 compared to those in intron 4, only canonical splicing with complete inclusion of exon 6 was observed in a corresponding mini-gene for exon 6 of *A3B* with 80 bp of flanking intronic sequences (E6$^{BPS+}$). Consistently, no canonical splicing was observed in the E6$^{BPS−}$ mini-gene, which lacks the BPS in intron 5 (Fig. 5e). Based on these results, we conclude that the skipping of *A3B* exon 5 is facilitated by weak BPS in intron 4. The magnitude of exon 5 skipping is likely to be determined by expression levels of some SFs that might differ between tissue types and disease conditions, including normal tissues vs. tumors.

**SF3B1 inhibitor pladienolide B promotes skipping of *A3B* exon 5 and reduces A3B1 production.** Efficient canonical splicing requires robust interaction of the spliceosomal machinery

with intronic BPS; thus, exon 5 skipping could result from a weak engagement of the spliceosome to BPS in intron 4 (Fig. 6a). Interaction between spliceosome and BPS is facilitated by the SF3b complex, with SF3B1 splicing factor as a core protein[32]. We hypothesized that interference with SF3B1 function, such as by treatment with a spliceosome inhibitor pladienolide B[32,33], would destabilize interaction of the SF3b complex at weak BPS within intron 4, resulting in enhanced exon 5 skipping, but may not affect exon 6 splicing that is regulated by strong BPS in intron 5 (Fig. 6b). We tested this hypothesis by evaluating splicing patterns of E5$^{BPS+}$ and E6$^{BPS+}$ mini-genes in cells treated with pladienolide B.

This treatment not only enhanced but also caused complete exon 5 skipping in a concentration-dependent manner while not affecting exon 6 splicing in the mini-gene models (Fig. 6c). Skipping of exon 5 in endogenous *A3B* was also increased by pladienolide B treatment, resulting in increased expression of *A3B3* mRNA and decreased expression of *A3B1* mRNA and A3B1 protein (Fig. 6d, e and Supplementary Fig. 8a). Importantly, the reduction of A3B1 protein caused by pladienolide B treatment was more prominent and consistent than the induction of *A3B3* expression. *A3B3* expression was much higher in cells treated with pladienolide B in the presence of digoxin, an NMD blocker[21] (Fig. 6f), supporting the NMD-based degradation of *A3B3* transcript (Fig. 1c) as a mechanism contributing to its poor detection on mRNA level and no detection on the protein level.

These experiments confirmed that exon 5 skipping is facilitated by weak BPS in intron 4 but is also dependent on the expression levels of SF3B1. The role of SF3B1 in this process was further confirmed by the effect of its siRNA-mediated knockdown in HT-1376 cells, which resulted in increased skipping of *A3B* exon 5 (Supplementary Fig. 8b). Thus, interfering with SF3B1 activity by

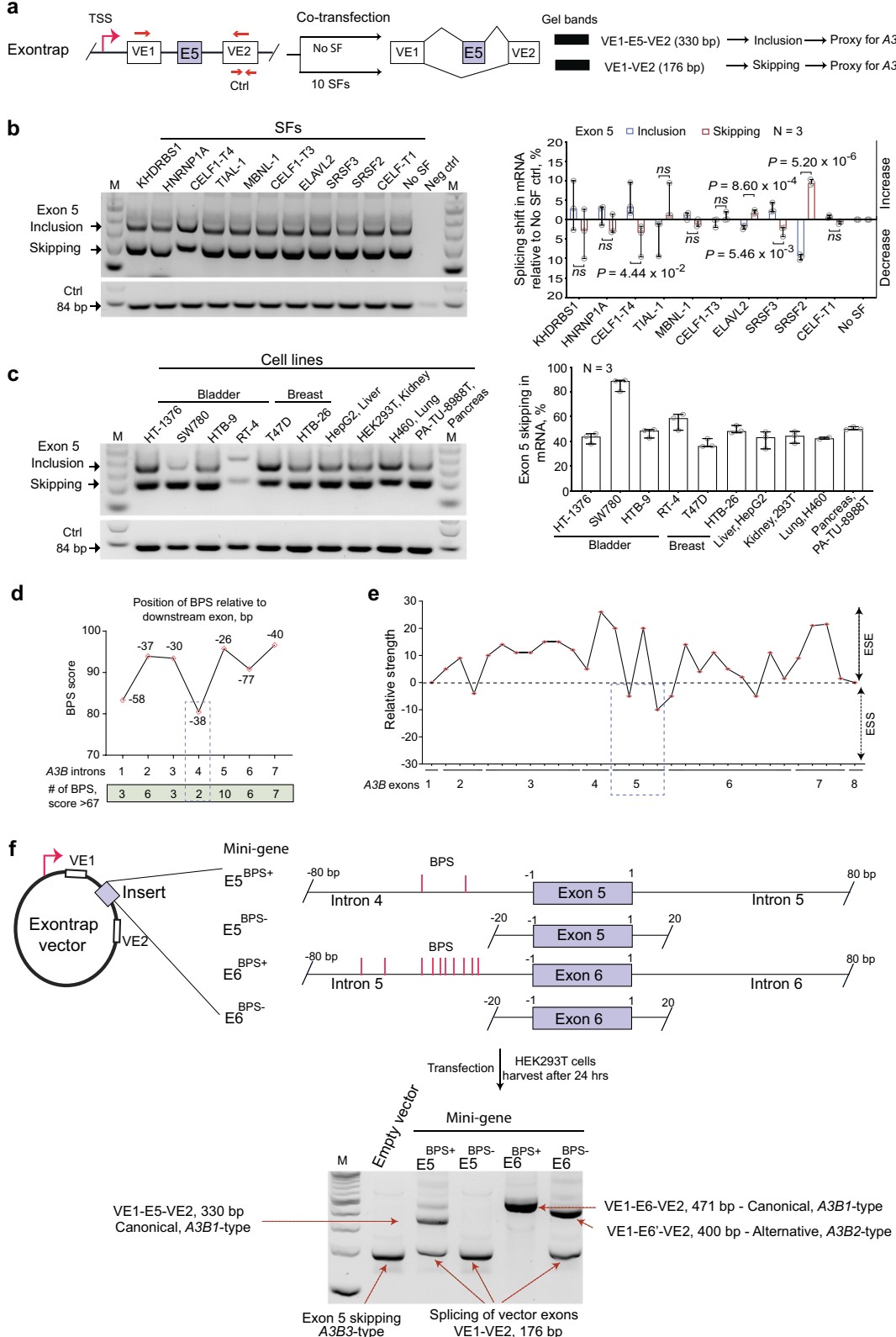

its inhibitors should increase *A3B* exon 5 skipping, leading to reduced levels of mutagenic A3B1.

Reducing the A3B1 levels increased by various endogenous and exogenous factors, including treatments, might restrict APOBEC mutagenesis. This restriction could be important for preventing tumor progression, recurrence, clonal evolution, and chemotherapy

resistance[34]. To explore this, we simultaneously treated cells with bleomycin, a chemotherapy drug, which induces *A3B* expression[1] and pladienolide B. Endogenous bleomycin-induced A3B caused deaminating (mutagenic) activity detectable in lysates of these cells, but this activity was completely blocked by pladienolide B (Fig. 6g). In the samples co-treated with bleomycin and pladienolide B,

**Fig. 5 The efficiency of exon 5 skipping depends on branch point sites in intron 4 of _A3B_. a** Exontrap mini-gene E5 as an experimental model for the analysis of AS of _A3B_ exon 5. **b** An agarose gel showing variable ratios of RT-PCR for splicing products of the E5 mini-gene at baseline (No SF control) and after co-transfection with 10 candidate SFs in HEK293T cells; negative control—untransfected cells; M—100-bp size marker; positive control—vector-specific amplicon of 84 bp. Bands from separate gels representing three biological replicates were quantified by densitometry, and the _A3B1/A3B3_ splicing shifts were calculated in relation to no SF control and plotted as a bar graph. Overexpression of SRSF2 significantly shifted splicing towards increased _A3B3_ expression (about 10%). **c** Splicing patterns of the E5 mini-gene analyzed by RT-PCR and agarose gel electrophoresis in 10 human cell lines. Bands from separate gels representing three biological replicates were quantified by densitometry to assess the skipping of _A3B_ exon 5, as shown in the bar graph. **d** Branch points site (BPS) prediction for all 7 introns of _A3B_. Shown are the BPS with the highest predicted scores for each intron, with indicated positions upstream of the corresponding exons and the numbers of predicted BPS in a 100 bp window. The two predicted BPS within intron 4 have the lowest scores of all _A3B_ introns. **e** Exon splicing enhancer (ESE) and exon splicing silencer (ESS) prediction within _A3B_ exons. Exon 5 has the strongest ESS motif of all _A3B_ exons. **f** Comparison of splicing patterns for _A3B_ exons 5 and 6 using mini-genes with (E5$^{BPS+}$ and E6$^{BPS+}$) and without (E5$^{BPS-}$ and E6$^{BPS-}$) predicted BPS. "E" refers to exon and "VE" to vector exon. Splicing between vector exons—VE1-VE2 is a proxy for _A3B3_ isoform created by exon 5 skipping. All bar graphs (**b**, **c**) show individual data points with error bars representing 95% confidence intervals for the medians, based on three biological replicates. _P_-values are for the two-sided unpaired Student's _t_-test.

A3B1 protein levels remained similar to untreated samples (Fig. 6h). These results suggest that increased A3B1 production can be prevented, such as by treatments that switch _A3B1_ mRNA to _A3B3_, such as with pladienolide B. The APOBEC-mediated mutagenic activity was also inhibited by pladienolide B in a cell-based cytosine deamination assay (Supplementary Fig. 9).

**Identification of splice-switching oligonucleotides to facilitate _A3B_ exon 5 skipping.** Splice-switching oligonucleotides (SSOs) can specifically and effectively modulate splicing events as therapeutic targets in various conditions, including cancer[35]. To this end, we designed SSOs to target _A3B_ splicing through two BPS in intron 4 (BPS-50 and BPS-38) and the 3′-splice site of exon 5 (3′ SS), and tested their ability to affect A3B exon 5 skipping in vitro (Fig. 7a). The SSO-BPS-50 had no effect and SSO-BPS-38 had a modest effect on exon 5 skipping (10%) (Fig. 7b), which could be due to RNA folding at these sites creating steric hindrance for binding, a common problem in identifying efficient SSOs[36]. However, the SSO-3′SS caused ~80% of _A3B_ exon 5 skipping (Fig. 7b), providing first insights into the utility of using SSOs for modulating splicing of _A3B_ through exon 5 skipping for controlling APOBEC-mediated mutagenesis (Fig. 7c).

**Discussion**
High activity of the A3A and A3B enzymes is mutagenic and genotoxic, as suggested by in vitro overexpression studies[37–40] and positive correlations between mRNA expression of genes encoding these enzymes and burden of APOBEC-signature mutations in tumors[1,8,25,41]. This potent mutagenic activity must be tightly regulated by multiple mechanisms to restrict the cell damage it may cause. Here, we showed that AS of APOBEC3s, and particularly of _A3B_, is one of several possible regulatory mechanisms that control the expression of the mutagenic A3B1 enzyme. We present proof-of-principle data suggesting that AS can be modulated to shift the _A3B_ balance from producing mutagenic to non-mutagenic isoforms.

Initially, we explored the functional consequences of splicing events within exon 2 in _A3A_ and exon 6 in _A3B_, both occurring via cryptic splicing sites, and another event, occurring through skipping of the entire exon 5 in _A3B_. In all these cases, AS resulted in a shift from the production of the canonical isoforms (_A3A1_ and _A3B1_) that encode mutagenic enzymes to corresponding alternative isoforms (_A3A1_, _A3B2_, and _A3B3_) that encode non-mutagenic enzymes. The expression of the canonical isoforms positively correlated with a load of APOBEC-signature mutations in TCGA tumors of different types. The isoform-level analysis also suggested that A3A1 and A3B1 might contribute to APOBEC mutagenesis in a cancer-type specific manner. Our previous finding that a common genetic variant identified

by a GWAS for bladder cancer risk was also associated with _A3B_ expression and APOBEC mutagenesis[1] nominated A3B as the primary mutagenic APOBEC in bladder tumors. This was further supported by the association of increased _A3B1_ expression with higher APOBEC mutation score and shorter progression-free survival in patients with non-muscle-invasive bladder cancer.

AS is regulated by a complex interplay between the core spliceosomal and alternative SFs[31] that bind cis-acting exonic and intronic elements. These interactions can depend on various tissue- and disease-specific environments[42]. Analysis of splicing patterns using our in vitro mini-gene Exontrap system identified splicing plasticity of _A3B_ exon 5, which we attributed to the weak BPS in intron 4 of this gene and variable levels of SF3B1 expression. Splicing of endogenous exon 5 also varied in cell lines of different tissue origin and was sensitive to changes in expression levels of some other SFs predicted to bind within this exon, such as SRSF2.

We observed that _A3B_ splicing events were more common in adjacent normal tissues compared to tumors of several types. This suggests that AS of _A3B_ is an intrinsic, tissue-specific regulatory mechanism rather than a result of general dysregulation of splicing machinery in tumors[43,44], manifested in the inactivation of tumor suppressor genes and generation of tumor-specific isoforms[45]. On the other hand, mutations in splicing or other regulatory factors that would affect splicing globally could also result in decreased splicing towards alternative _A3B2_ and _A3B3_ isoforms, failing to prevent shunting of _A3B1_ mRNA and consequently lead to increased APOBEC mutagenesis. Thus, AS of _A3B_ might be a natural mechanism restricting the expression of mutagenic isoforms in some conditions, such as in normal tissues.

The observed splicing plasticity of _A3B_ mRNA might represent an adaptive biological mechanism of tuning down the excessive effects of mutagenic APOBEC3 proteins to safeguard the cells from the genotoxic activity of these enzymes. A similar role has been proposed for several SFs, including SRSF2, which affected _A3B_ exon 5 splicing in our experiments. These SFs regulate the expression of DNA repair proteins to protect the genome from DNA damage and the toxic effects of mutagens[46–48]. Splicing re-routing, such as by exclusion of exon 5 in _A3B_, followed by elimination of the alternative frame-shifted _A3B3_ transcript by NMD might be a mechanism to tweak APOBEC mutagenesis in specific conditions. The entire functional role of _A3B3_ could be just to use up some pre-mRNA that otherwise would be used to produce the mutagenic A3B1 enzyme and then get degraded by NMD. Thus, the NMD-targeted _A3B3_ transcript not producing a functional protein might still play an important role in the regulation of APOBEC mutagenesis, regardless of its very low residual expression levels observed in tumors and tissues.

A recent study has analyzed mutational signatures in a large set of cell lines and suggested that the initiators of APOBEC

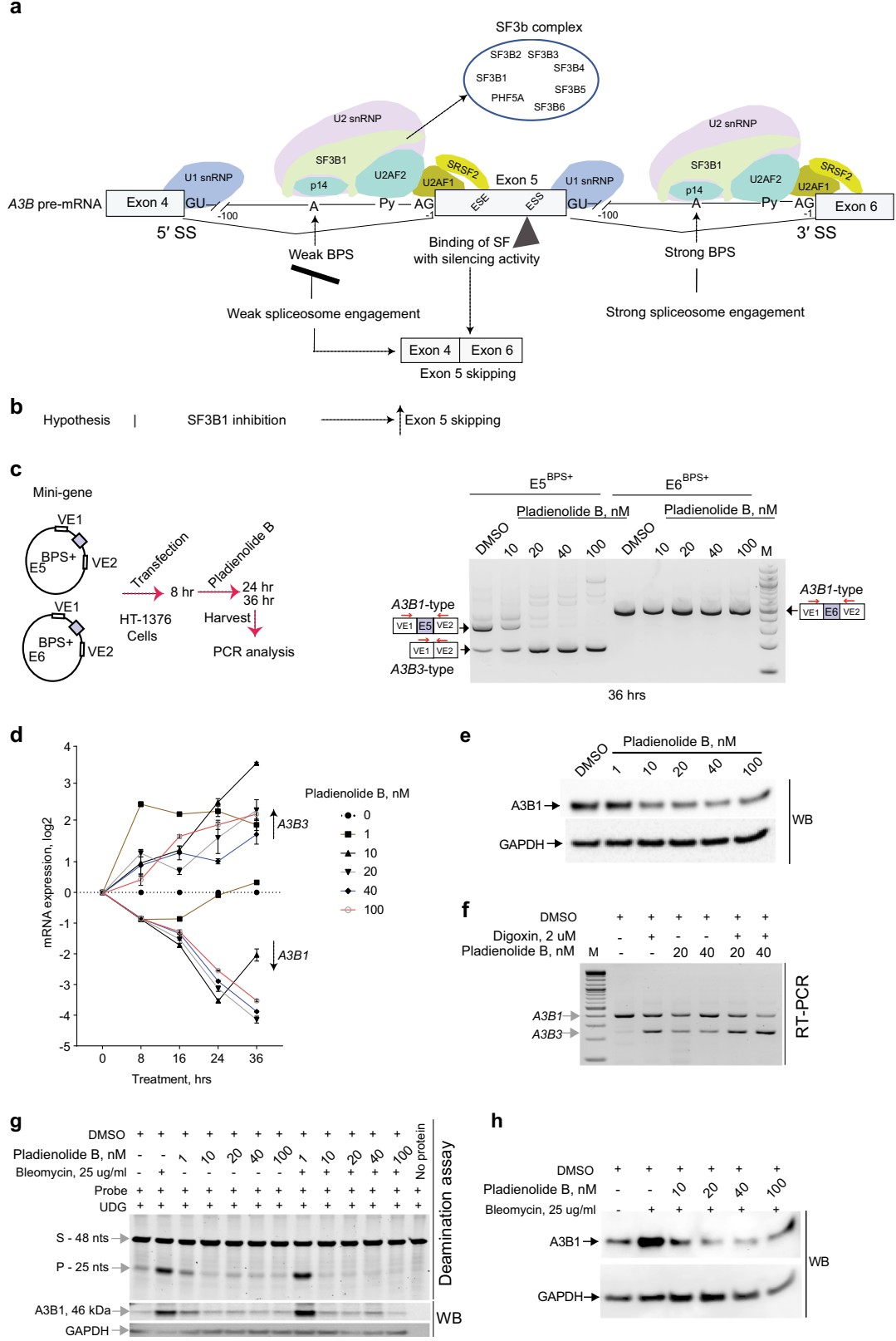

mutagenesis in vitro are cell-intrinsic factors with continuous but intermittent activity[26]. As the authors did not observe correlations between APOBEC mutagenesis and expression of *APOBEC3* genes, they suggested that these initiators may include modulators such as the availability of single-stranded DNA

(ssDNA) substrate, etc. Based on our observations that *A3B1* and not *A3A1* is expressed in bladder cancer cell lines, which have a high load of APOBEC-signature mutations, it is likely that at least in these cell lines, episodic APOBEC-mediated mutagenesis is caused by the A3B1 activity.

**Fig. 6 Modulation of *A3B* exon 5 skipping by an SF3B1 inhibitor pladienolide B. a** Schematic representation of the *A3B* splicing module (exon 4–exon 6) featuring interactions between *trans*-factors of the spliceosomal machinery with splicing *cis*-elements: splice sites (SS), branch point sites (BPS), polypyrimidine tract (Py), and exonic splicing enhancers/silencers (ESE)/(ESS); possible outcomes of *A3B* exon 5 splicing resulting from interactions between SF3B1 and other spliceosomal proteins with weak versus strong BPS. **b** Schematic representation of the hypothesis depicting that inhibition of SF3B1 is expected to result in skipping of *A3B* exon 5 and downregulation of A3B1 protein. **c** PCR analysis of splicing patterns of *A3B* exons 5 and 6 using Exontrap mini-genes E5^BPS+ and E6^BPS+ in HT-1376 cells treated for 36 h with increasing concentrations of pladienolide B. Exon 5 skipping is increased in the presence of the SF3B1 inhibitor, while splicing of exon 6 is not affected. **d** Results of qRT–PCR in HT-1376 cells treated with increasing concentrations of pladienolide B for indicated time points show splicing shift from *A3B1* to *A3B3*. Shown are means ± SD of biological triplicates. **e** Western blot protein analysis of HT-1376 cells treated with increasing concentrations of pladienolide B for 36 h showed downregulation of A3B1 protein, while expression of endogenous A3B3 protein was not detected. **f** RT-PCR analysis of exon 5 splicing representing *A3B1* and *A3B3* transcripts in HT-1376 cells treated with DMSO (vehicle), digoxin (an NMD inhibitor), pladienolide B alone, and pladienolide B with digoxin. *A3B3* mRNA levels are increased in digoxin-treated cells due to its inhibition of NMD. **g** Deamination assays using whole-cell extracts from HT-1376 cells treated with bleomycin and pladienolide B. Equal amounts of total protein were used for each reaction based on densitometry pre-assessment of GAPDH protein levels in each sample. Deamination was observed in reactions with protein lysate of HT-1376 cells treated with bleomycin alone or bleomycin plus 1 nM of pladienolide B. Higher concentrations of pladienolide B (10, 20, 40, and 100 nM) reduced A3B1 levels and prevented deamination of the ssDNA probe in bleomycin-treated conditions. A3B1 was barely detectable in DMSO and pladienolide B only conditions because of low total protein input in reactions (2 μg of total protein lysate). **h** Separate experiment showing that pladienolide B (10, 20, 40, and 100 nM) suppresses induction of A3B by bleomycin. Western blot (WB) analysis with an anti-A3B1 antibody shows the amounts of A3B1 in corresponding reactions. Each lane was loaded with equal amounts of the whole-cell lysate (15 μg of protein/lane). Shown are representative results of one of the three independent experiments.

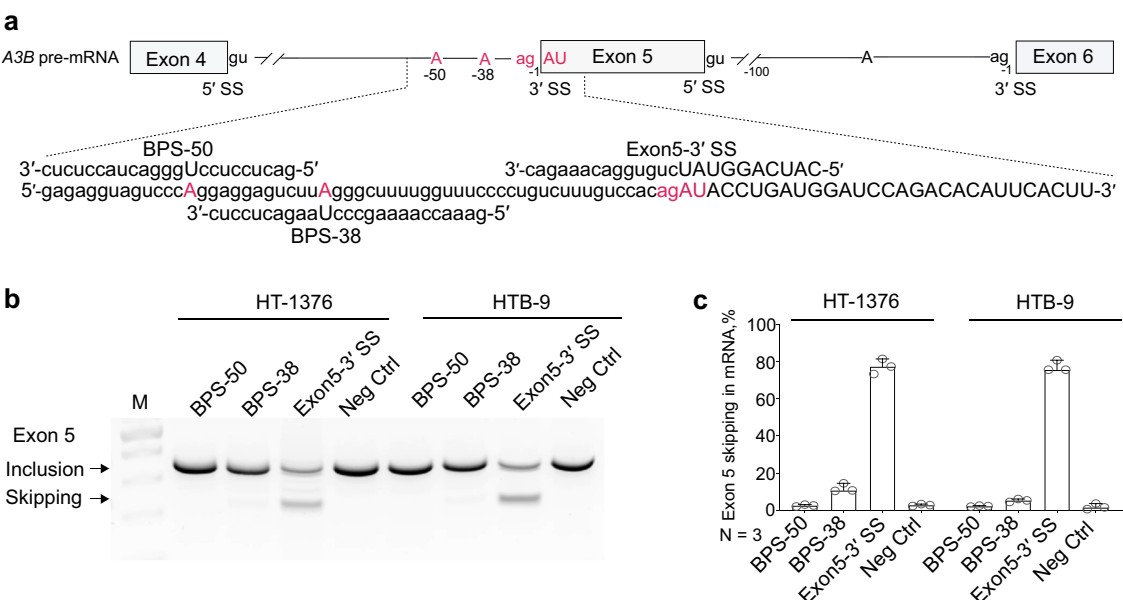

**Fig. 7 Splice-switching oligonucleotides (SSOs) for inducing *A3B* exon 5 skipping. a** Schematic representation of the *A3B* splicing module that includes exons 4–6. The splice sites (SS), branch point sites (BPS), and SSOs for intron 4 BPS (positions −50 and −38 bp) and 3′ SS of exon 5 are depicted. **b** A representative agarose gel showing RT-PCR products of exon 5 inclusion and skipping in two bladder cancer cell lines (HT-1376 and HTB-9) that were nucleofected with 2 nM of SSOs. **c** Bar graphs based on image analysis of **b**, showing the % of *A3B* exon 5 skipping. Error bars represent 95% confidence intervals for the medians, based on three biological replicates. SSO-3′SS resulted in exon 5 skipping in ~80% of A3B mRNA.

Therapeutic targeting of AS to treat cancer is a rapidly developing field[49] with several types of SF3B1 inhibitors, such as E7107[50,51], an analog of pladienolide B, and H3B-8800[52] being evaluated for acute myeloid leukemia (AML) and myelodysplastic syndrome (MDS)[33,49,52]. Because tumor cells depend on wild-type SFs for survival, these drugs are particularly effective in killing cancer cells with already impaired splicing machinery due to mutations in SFs, such as SRSF2 and SF3B1[52]. Mutations in these and other SFs have been reported in many solid tumors, including bladder cancer[53] and these tumors tend to have more APOBEC-signature mutations (Supplementary Fig. 10). Our results suggest that SF3B1 inhibitors and other tools affecting alternative splicing might be tested to eliminate cancer cells with mutations in SFs and also to control APOBEC mutagenesis in

clinically relevant conditions. It is interesting to note that *FGFR3*-S249C, the most common somatic mutation found in the highly recurrent non-muscle-invasive bladder cancer, is likely caused by APOBEC mutagenic activity[54]. As we found A3B1 the primary mutagenic APOBEC enzyme in bladder tumors, enhancing *A3B* exon 5 skipping might help to restrict APOBEC-mediated mutagenesis, including *FGFR3*-S249C mutation in bladder cancer, as well as prevent tumor progression and recurrence, clonal evolution, and resistance to chemotherapy (Fig. 8). In addition to using SF3B1 inhibitors, we also demonstrate the utility of SSOs for inducing exon 5 skipping and decreasing the expression of mutagenic A3B isoform, possibly opening up avenues for targeting APOBEC-mediated mutagenesis by oligo-based drugs[55]. Using SSOs to increase the skipping of exon 5 of *A3B* would

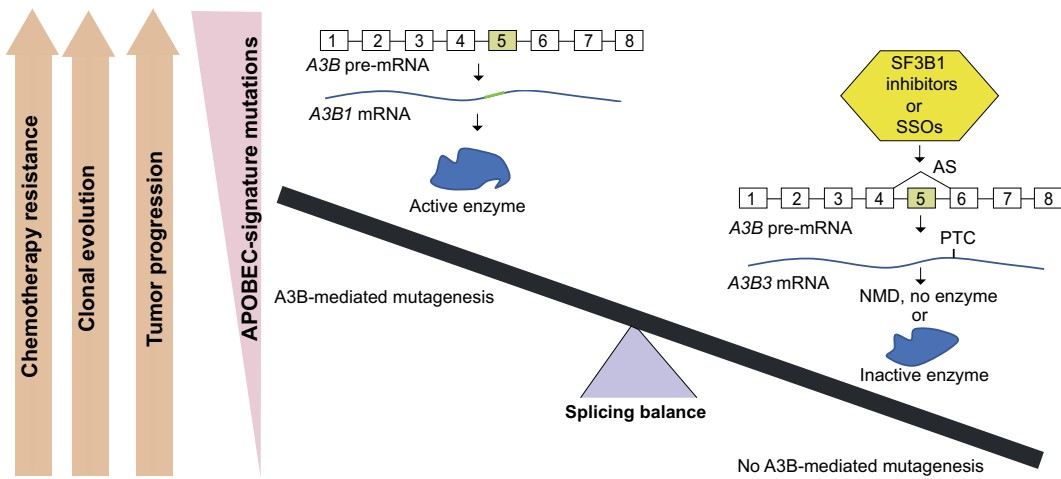

**Fig. 8 The proposed model to target alternative splicing of *A3B* for modulating APOBEC mutagenesis.** Canonical splicing of *A3B* pre-mRNA generates the mutagenic A3B1 enzyme that causes APOBEC mutagenesis and fuels tumor progression, recurrence, clonal evolution, and chemotherapy resistance. Alternative splicing of *A3B*, and, specifically, skipping of exon 5, generates *A3B3* isoform with a premature termination codon (PTC). Some of this aberrant transcript is degraded by nonsense-mediated decay (NMD), and the residual transcript encodes catalytically inactive protein isoform. Therapeutic enhancing of exon 5 skipping by SF3B1 inhibitors or other tools affecting alternative splicing, such as splicing-switching oligonucleotides (SSOs), may restrict and even prevent A3B-mediated mutagenesis in clinically relevant conditions.

result in natural NMD-based elimination of the non-functional A3B3 isoform produced in this process. Targeting any other gene region of the highly homologous *APOBEC3* gene family (Supplementary Note 1) might be more challenging and cause off-target effects.

In conclusion, our study showed that AS of *A3B*, which creates non-mutagenic isoforms, represents an intrinsic regulatory mechanism that keeps in check the expression of the mutagenic A3B1 enzyme, modulates APOBEC-mediated mutagenesis in different disease-, tissue- and environment-specific conditions, and can be therapeutically targeted. Additional functional studies will be needed to better understand the consequences of this modulation for clinical outcomes.

## Methods

**Data acquisition and processing of TCGA samples**. The mutation dataset for all TCGA cancer types was downloaded from the Broad GDAC Firehose in October 2016. For analysis of APOBEC-signature mutations, we used the variable "APO-BEC mutation load minimum estimate", which represents APOBEC mutation pattern[8]. mRNA expression of *A3A* and *A3B* genes was analyzed in 11,058 TCGA samples (10,328 tumors and 730 adjacent normal tissues) based on RNA-seq BAM slices generated using workflow https://docs.gdc.cancer.gov/API/Users_Guide/BAM_Slicing/ through the NCI Genomics Data Commons (GDC) portal accessed in August 2019. RNA-seq BAM files for cell lines from the Cancer Cell Line Encyclopedia (CCLE) were acquired from the GDC legacy archive (https://portal.gdc.cancer.gov/legacy-archive/search/f) and Stringtie-2.1. 4 was used for whole-transcriptome level gene expression quantification using hg19 as the reference.

**Estimation of *A3A* and *A3B* RNA-seq read counts for exons and exon–exon junctions**. For estimation of sequencing reads corresponding to all exons as well as canonical and alternatively spliced exon–exon junctions, RNA-seq BAM slices were processed through R package ASpli version 1.5.1. We used strict ASpli pipeline settings, requiring a minimum of 10 nucleotides of perfect match covering exon–exon junctions.

**Correlation analysis of mRNA expression and APOBEC mutagenesis**. Expression levels of the mutagenic isoforms *A3A1* and *A3B1* were calculated based on their RNA-seq exon–exon junction read counts. The counts of sequencing reads for E1–E2 junctions (for A3A1) and E5–E6 junctions (for A3B1) and APOBEC mutation load minimum estimate counts were Log10-transformed (after adding 1 to all raw values). The Spearman correlation analysis and data plotting were performed using R packages for six cancer types with ≥10% of samples with APOBEC-signature mutations. SKCM was excluded from the analysis because mutations induced by APOBEC in melanoma likely overlap with mutations induced by UV[56].

**Analysis of AS levels in paired tumor and adjacent normal tissue samples**. The analysis was performed in 17 cancer types with ≥5 tumor-normal pairs. Splicing ratios were calculated as PSI based on RNA-seq reads using ASpli pipeline, as was previously described[28] and statistical significance between PSI in tumors vs. normal tissues was evaluated by non-parametric Wilcoxon matched-pairs signed-rank test.

**RNA-sequencing of SeV-infected T47D cells**. Total RNA extracted from the T47D breast cancer cells infected or not infected (control) with Sendai Virus (SeV)[1] was used for paired-end RNA-seq on HiSeq 2500 (Illumina) in biological duplicates (SRA accession number: PRJNA512015). The library was prepared with KAPA Stranded RNA-seq Kit with RiboErase (Roche). RNA-seq reads (120 bp) were filtered and aligned with STAR alignment tool[57] using the GRChg37/hg19 genome assembly and visualized using the Integrative Genomics Viewer (IGV). The RNA-seq reads aligned to *A3A* were BAM-sliced with SAMtools and then re-aligned to the reference genome by STAR to identify cross-alignment with *A3B*.

**Bioinformatics analysis of the A3A and A3B protein isoforms**. Protein sequences of the APOBEC3 isoforms A3A1 (UCSC ID uc003awn.2), A3A2 (UCSC ID uc011aob.1), A3B1 (UCSC ID uc003awo.1), A3B2 (UCSC ID uc003awp.1), and A3B3 (UCSC ID uc003awq.1) were downloaded from the UCSC genome browser. Multi-sequence alignments were generated using the web-based tool Clustal Omega[58].

**Cell lines**. Cell lines—embryonic kidney HEK293T, bladder cancer cell lines - HT-1376, RT-4, HTB-9, and SW-780, breast cancer cell lines—MDA-MB-231 (HTB-26), and T47D (HTB-133), hepatocellular carcinoma HepG2, cervical adeno-carcinoma HeLa and lung carcinoma H460—were purchased from the American Type Culture Collection (ATCC) and maintained per ATCC recommendations. A pancreatic cancer cell line PA-TU-8998T was purchased from Leibniz Institute DSMZ-German Collection of Microorganisms and Cell Cultures (DSMZ Scientific). No commonly misidentified cell lines were used in this project. If used longer than for a year after initial purchase, cell lines were authenticated by the Cancer Genomics Research Laboratory of DCEG/NCI by genotyping a panel of micro-satellite markers (Identifiler kit, ThermoFisher Scientific). All cell lines were tested bi-monthly for mycoplasma contamination using the MycoAlert Mycoplasma Detection kit (Lonza).

**Analysis in non-muscle-invasive bladder tumors from the UROMOL study**. A set of low-stage (Ta and T1) bladder tumors representing non-muscle-invasive bladder cancer (NMIBC) has been described[27]. Mutations were scored based on RNA-seq data and used for deconvolution into mutational signatures S1–S6, with S3 corresponding to the APOBEC-signature mutations[27]. The FASTQ files for RNA-seq data were aligned with STAR and BAM-sliced to include *A3A* and *A3B* genes, followed by estimation of all exon and exon–exon junction reads using ASpli R package, similar to the analysis in TCGA samples. We performed a Spearman correlation analysis of log10-transformed read counts for *A3A1* and *A3B1* with

APOBEC-signature mutation score (S3). Progression-free survival analysis was performed based on the expression of *A3A1* and *A3B1* mutagenic isoforms with samples divided into three groups: "No"—samples with undetectable expression (0 value); the remaining samples were split into two groups based on the expression below and above the median as "Low" and "High" groups, respectively. The Cox-regression models were adjusted for sex, age, and tumor stage.

**Analysis of additional bladder tumor and adjacent normal tissue samples**. The panel of muscle-invasive bladder tumors ($n = 42$) and adjacent normal ($n = 32$) tissue samples has been described[59]. For each of these samples, cDNA was synthesized from 250 ng of total RNA per 20 µl reactions using the RT2 First-Strand cDNA kit and random hexamers (Qiagen). For detection of splicing events between *A3B* exons 4 and 6, we performed PCR with primers: F_ex4: 5′-GCCTTGGTA-CAAATTCGATGA-3′ and R_ex6: 5′-TGTGTTCTCCTGAAGGAACG-3′, with cDNA input corresponding to 30 ng of total RNA per 25 µl reactions using AmpliTaq Gold™ 360 Master Mix (ThermoFisher Scientific). PCR-amplified products were resolved on 2% agarose gel, and each distinct PCR product was cut, purified, and Sanger-sequenced. For quantification of *A3B1* and *A3B3* splicing products, the cDNA input corresponding to 10 ng of total RNA per reaction was used, as previously described[1], and subjected to qRT–PCR using isoform-specific TaqMan assays (Supplementary Note 4). Expression of *A3B1* and *A3B3* was normalized by the expression of endogenous controls *GAPDH* (assay 4326317E) and *PPIA* (assay 4326316E). Western blotting for A3B1 protein and GAPDH (loading control) was performed as described in Supplementary Note 2.

**Generation and partial purification of the recombinant A3A and A3B protein isoforms**. Expression constructs for the C-terminally Myc-DDK tagged canonical isoforms A3A1 (NM_145699) and A3B1 (NM_004900) cloned in the pCMV6 vector were purchased from OriGene (Rockville, MD). Open reading frames for the C-terminally Myc-DDK tagged alternative isoforms A3A2, A3B2, and A3B3 (Supplementary Table 1) were synthesized (ThermoFisher Scientific) and cloned into BamHI/XbaI restriction sites of the pcDNA3.1(+) vector. The HEK293T cells ($4 \times 10^6$ cells/20 ml) were seeded in 175 cm² flasks (Corning) and transiently transfected with plasmids after 24 h at 75% confluency using Lipofectamine 3000 (ThermoFisher Scientific). Cells were harvested 24 h post-transfection and proteins were purified with c-Myc tagged Protein Mild Purification Kit (MBL, Japan) and eluted with 20 µl of 1 mg/ml Myc peptide provided with the kit. The concentration of the total eluted protein, which included both purified protein and Myc peptide, was estimated using a BCA protein assay (ThermoFisher Scientific). For evaluating protein purity and enrichment, ~25 µg of total protein was resolved on 4–12% Tris-glycine SDS polyacrylamide gel (Life Technologies) and used for Western blot analysis. Densitometry analysis of Western blots showed at least 10-fold enrichment of all eluted isoforms compared to whole-cell lysates (Supplementary Note 3). All candidate antibodies were tested for the detection of A3A and A3B protein isoforms after overexpression of corresponding expression constructs in cell lines (Supplementary Note 2). Detection was done using a HyGLO chemiluminescent HRP antibody detection reagent (Denville Scientific Inc).

**Cytosine deamination assays with recombinant proteins**. Deamination activity of the recombinant A3A and A3B protein isoforms was tested as previously described[60]. Specifically, reactions were carried out in 10 µl of deamination buffer containing 10 mM Tris/HCl, pH 7.5, 50 mM NaCl, 1 mM DTT, 0.25–1 µg of partially purified A3A and A3B proteins, and 1–5 pM of single-stranded DNA substrate—the target probe 5′-5Alexa488N/(ATA)₈TCC (ATA)₇-3′, or a positive control probe 5′-5Alexa488N/(ATA)₈TUU (ATA)₇-3′ (Invitrogen). Reactions were incubated in a water bath at 37 °C for 2 h, treated with the Uracil DNA Glycosylase (UDG) for 40 min at 37 °C, and then with 0.6 N NaOH for 20 min at 37 °C. Final products were mixed with 2x RNA loading dye (ThermoFisher Scientific) and heated at 95 °C for 2–3 min. Of the final 40 µl reaction volume, one set of 14 µl aliquots was resolved on 15% TBE-urea polyacrylamide gel (Life Technologies) at 150 V for 1 h and 20 min at room temperature in 1× TBE buffer. Gels were imaged with Gel Doc (Bio-Rad) using Alexa 488 fluorescence settings. Another set of 14 µl aliquots from the same reactions was used for the detection of APOBEC3 proteins by Western blotting. Concentrations of eluted proteins were estimated based on densitometry of Western blots. For competition assays, the amounts of mutagenic isoforms and total reaction volumes were kept constant while the amounts of non-mutagenic isoforms were increased. Proteins extracted from the lysates of untransfected cells were used as a negative control to account for inhibition caused by non-specific endogenous proteins. In the 1:1 control competition reaction, the non-mutagenic isoform was replaced by an equal amount of the negative control protein.

Deamination activity of endogenous A3B1 in the presence of pladienolide B was evaluated using a previously described protocol[14] with some modifications. Briefly, HT-1376 cells, treated with bleomycin (25 µg/ml) alone, pladienolide B alone (1, 10, 20, 40, 100 nM) and with combined bleomycin and pladienolide B, were harvested after 48 h in HED buffer (25 mM HEPES, 5 mM EDTA, 10% glycerol, 1 mM DTT and protease inhibitor). The protein concentrations were determined based on a densitometric assessment of GAPDH by Western blot from 7 µl of total

lysate for each condition. Each 20 µl deamination reaction contained equal amounts of total protein in 15 µl (adjusted with H₂O), 1 µl (10 pM) of single-stranded DNA substrate–probe 5′-5Alexa488N/(ATA)₈TCC (ATA)₇-3′, 2 µl 10× UDG buffer (ThermoFisher Scientific), 1 µl (1 U/µl) UDG and 1 µl RNaseA (100 mg/ml, Qiagen) and reactions were incubated at 37 °C for 3 h. Subsequently, 100 mM NaOH was added to each reaction, and the samples were then incubated at 37 °C for 30 min to cleave the abasic sites. Final products were mixed with 2× RNA loading dye (ThermoFisher Scientific) and heated at 95 °C for 2–3 min, and reactions were resolved on a 15% urea-TBE gel and imaged as described above.

**HIV-1 infectivity inhibition assays**. The activity of the A3B protein isoforms was evaluated with single-cycle infection assays for HIV-1 restriction as has been described for APOBEC3G (A3G)[22–24]. Briefly, a G/B-A3B1 plasmid was constructed starting from the previously engineered A3B plasmid[22] by replacing 63 aa at the N-terminus with a similar region of A3G. This replacement increases the packaging of A3B protein into HIV-1 viral particles, which is necessary for the ability of A3B to inhibit HIV-1 infectivity[22,24]. This system is suitable for analysis of the activity of A3B, which like A3G, has two cytidine deaminase domains. However, this approach is not appropriate for evaluating the activity of A3A as it contains only one cytidine deaminase domain, and A3G/A3A swap would not be compatible. Fusion constructs G/B-A3B-V1, G/B-A3B-V2, G/B-A3B-V3, were generated to represent three A3B expression constructs, all with C-terminal hemagglutinin epitope tags (3X-HA). All plasmids were verified by Sanger sequencing. Because the A3B plasmids were from three sources, we found two protein-changing single point variations that might be functionally relevant (Supplementary Fig. 3). To generate the virus for infection, HEK293T cells ($4 \times 10^5$ cells/6-well dish) were transfected using LT1 reagent (Mirus Bio) with HDV-EGFP (1 ug), pHCMV-G (0.25 ug), and variable concentrations of plasmids (0, 680, and 1200 ng), individually or in combinations. The virus was harvested 48 h post-infection, filtered with 0.45-um-pore filters, and stored at −80 °C. Capsid p24 measurements were determined using the HIV-1 p24 capsid (CA) ELISA Kit (XpressBio). Normalized p24 CA amounts were used to infect TZM-bl cells containing HIV-1 Tat-inducible luciferase reporter gene in a 96-well plate (4000 cells/well). Luciferase activity was measured 48 h after infection, using a 96-well luminometer (LUMIstar Galaxy, BMG LABTECH). For some experiments, portions of the viral supernatant were spun through a 20% sucrose cushion (15,000 rpm, 2 h, 4 °C, in a Sorvall WX80 + ultracentrifuge), concentrated 10-fold, and used in experiments to determine virion encapsidation of APOBEC3 proteins by western blotting analysis as previously described[23].

**Evaluation of *A3B3* mRNA degradation by nonsense-mediated decay (NMD)**. The HT-1376 cells were treated with DMSO (vehicle) or with 2, 5, and 10 µM of digoxin (Sigma), an NMD inhibitor[21] dissolved in DMSO. The cells were harvested after 16 and 24 h of treatment; total RNA was isolated with an RNeasy kit with on-column DNase I treatment (Qiagen), and RNA quantity and quality were analyzed with NanoDrop 8000 (Thermo Scientific). After an additional DNA removal step, cDNA for each sample was prepared from equal amounts of total RNA, using the RT² First-Strand cDNA kit and random hexamers (Qiagen). Expression was measured in the same cDNA with TaqMan expression assays (all from Thermo-Fisher Scientific) for endogenous controls *GAPDH* (assay 4326317E) and *PPIA* (assay 4326316E), and positive control *ATF4* (assay Hs00909569_g1), which is induced by NMD inhibition[61]; custom assays were used for detection of *A3B1*, *A3B3*, and *A3B1/A3B2* combined (Supplementary Note 4). Experiments were performed in biological triplicates per condition and expression was measured in four technical replicates on QuantStudio 7 (Life Technologies) using TaqMan Gene Expression buffer (Life Technologies). Water and genomic DNA were used as negative controls for all assays. Expression was measured as $C_t$ values (PCR cycle at detection threshold) and calculated as fold change using the $2^{-(\Delta\Delta Ct)}$ method in relation to control (untreated) groups of samples.

**Bioinformatics analysis of *A3B* splicing *cis*-elements and SF binding sites**. Exons and 100 bp of flanking intronic sequences were used for the prediction of exonic splicing enhancer (ESE)/silencer (ESS) motifs and branch point sites (BPS) using the Human Splicing Finder (HSF, www.umd.be/HSF3/)[62]. Per HSF guidelines, BPS with scores above 67 were considered high-confidence; the strength of ESE and ESS was evaluated based on the relative ESE/ESS ratio. SF binding sites were predicted using SFmap[63] and SpliceAid2[64] (Supplementary Table 3).

**Exontrap analysis of alternatively spliced exons of *A3A* and *A3B* genes**. Exon 2 of *A3A* and exons 5 and 6 of *A3B* with either 20 or 80 bp of flanking intronic sequences were custom-synthesized (ThermoFisher Scientific) and cloned in sense orientation using XhoI and NotI restriction sites of Exontrap vector pET01 (MoBiTec) to generate mini-genes that were validated by Sanger sequencing. The HEK293T cells were seeded in a 96-well plate at a cell density of $1.5 \times 10^4$ and transfected the next day with 100 ng of mini-genes using Lipofectamine 3000 transfection reagent (Invitrogen), in 4 biological replicates. Cells were harvested 48 h post-transfection, and total RNA was extracted with QIACube using RNeasy kit with on-column DNase I treatment (Qiagen). For each sample, 0.5–1 µg of total

RNA was converted into cDNA with SuperScript III reverse transcriptase (Invitrogen) and a vector-specific primer: 5′-AGGGGTGGACAGGGTAGTG-3′. Samples were diluted with water, and cDNA corresponding to 1.5 ng of RNA input was used for each qRT–PCR reaction. Splicing products of each mini-gene were amplified using a common primer pair F: 5′-CACCTTTGTGGTTCTCACTTGG-3′ and R: 5′-AGCACTGATCCACGATGCC-3′, corresponding to vector exons V1 and V2 (Figs. 4, 5 and Supplementary Fig. 6). An assay with primers F: 5′-CCGT GACCTTCAGACCTTGG-3′ and R: 5′-AGAGAGCAGATGCTGGTGCA-3′ targeting Exontrap vector exon V2 was used as a control. All PCR-amplified splicing products were resolved by agarose gel electrophoresis. Specific bands were cut out from the gel, purified, and validated by Sanger sequencing. Co-transfection of E5 mini-gene with 10 SFs was performed in HEK293T cells for 48 h, followed by RNA extraction and analysis of splicing patterns. Similar analyses were performed for four select SFs (SRSF2, SRSF3, CELF1, and ELAVL2) in SW-780, HT-1376, and HTB-9 cells. The E5 construct was also transfected into a panel of 10 cell lines for 24 h, and splicing analysis was performed as described above.

**Modulation of *A3B* exon 5 splicing**. Cell lines (HT-1376, HTB-9, and HeLa) were grown in 12-well plates at a density of $1.5 \times 10^5$ cells/well and treated with DMSO or 1, 10, 20, 40, and 100 nM pladienolide B (Santa Cruz, sc-391691) reconstituted in DMSO, in biological triplicates per condition. Cells were harvested after 8, 16, 24, and 36 h of treatment separately for RNA and protein analysis. cDNA was synthesized from 300 ng of RNA in a 20 µl reaction using the RT² First-Strand cDNA kit and random hexamers (Qiagen). cDNA corresponding to 30 ng of total RNA per reaction was used for PCR performed with a primer pair to amplify different splicing products of *A3B*: Forward primer on exon 4: 5′-GCCTTGGTAC AAATTCGATGA-3′, Reverse primer on exon 6: 5′-TGTGTTCTCCTGAAGGAA CG-3′. PCR-amplified splicing products, *A3B1* (397 bp), *A3B2* (322 bp), and *A3B3* (243 bp), were resolved by agarose gel electrophoresis, and specific bands were cut out from the gel, purified, and validated by Sanger sequencing. Quantification of *A3B1* and *A3B3* splicing products was performed with qRT–PCR using isoform-specific TaqMan assays (Supplementary Note 4) and cDNA corresponding to 10 ng of total RNA per reaction, as previously described[1]. Expression of *A3B1* and *A3B3* was normalized by the expression of endogenous controls *GAPDH* (assay 4326317E) and *PPIA* (assay 4326316E). Water and genomic DNA were used as negative controls for all assays. Expression was measured as $C_t$ values (PCR cycle at detection threshold) and calculated as $\Delta\Delta Ct$ in relation to control (DMSO) groups of samples. For protein analysis, whole-cell extracts were harvested in RIPA buffer with proteinase inhibitor (ThermoFisher Scientific) and used for Western blotting as described in Supplementary Note 2. Shown are representative results of one of three independent experiments.

**Cell-based cytosine deamination assay**. To determine base substitution rates, we used an ssDNA oligo (Supplementary Fig. 9a), modified based on a previously reported oligo[65]. HT-1376 cells were nucleofected with 100 pmol of oligo alone or together with A3B1 plasmid (program CM-130) with SF Cell Line 4D X Kit L (V4XC-2024) on Lonza 4D-Nucleofector. Nucleofected cells were then plated in a six-well plate and after 8 h, cells transfected only with oligo were treated with DMSO or pladienolide B for 64 h. Cells were then lysed with QuickExtract DNA Extraction Solution (Lucigen) and PCR-amplified using primers: Forward, 5′-TGA TGATGTGAGTGGTGGATGA-3′; Reverse, 5′- TCATCAACACCTACCACACA C-3′. PCR products were gel-purified with a DNA gel extraction kit (QIAquick Gel Extraction Kit, Qiagen) and used for library preparation with TruSeq/ChIP-Seq reagents (Illumina), to generate 75 bp paired-end sequencing reads. Samples were barcoded, multiplexed, and subjected to deep sequencing on the Illumina MiSeq instrument. The FASTQ files were aligned with custom reference (sequence of ssDNA oligo) using the BWA-MEM algorithm and then indexed by SAMtools.

**siRNA knockdown of SF3B1**. The HT-1376 cells were transfected with scrambled siRNA (#1022076), SF3B1 siRNA-1 (SI00715932), or siRNA-2 (SI04154647), all from Qiagen, using Lipofectamine RNAiMAX Reagent (ThermoFisher). Cells were harvested after 36 h for RNA and protein and analyzed for *A3B* exon 5 skipping by PCR as described above. SF3B1 knockdown was confirmed by Western blot with an anti-SF3B1 antibody (Abcam, ab172634, 1:1000 dilution) and GAPDH control, as described above.

**Modulation of A3B exon 5 skipping by splice-switching oligonucleotides (SSOs)**. We designed SSOs targeting two predicted BPS within intron 4 of A3B (at positions -38 and -50 bp) and the 3′ splice site (3′SS) of exon 5 (Fig. 7a). The SSOs were 2-OMe-substituted phosphorothioate RNA (Integrated DNA Technologies): BPS-50 (5-GACUCCUCCUGGGGACUACCUCUC-3), BPS-38 (5-GAAACCAA AAGCCCUAAGACUCCUC-3), 3′SS (5-CAUCAGGUAUCUGUGGACAAAGA C-3), and negative control (5-AGUAGGAAAAGGGAUAUAUGAUGGAAAU-3). HT-1376 and HTB-9 cells were nucleofected with 2 nM of SSOs (program CM-130) using SF Cell Line 4D X Kit L (V4XC-2024) on Lonza 4D-Nucleofector. Nucleofected cells were then plated in a 12-well plate and harvested after 48 h for RNA extraction and analysis of A3B exon 5 skipping by PCR as described above.

**Computational analysis**. All data processing and analyses were performed using R package versions (3.2.4-3.4.0), SPSS version 25, and NIH High-Performance Computing Biowulf cluster.

**URLs**. Firehose Broad GDAC: https://gdac.broadinstitute.org/; Firebrowse: http://firebrowse.org/#; The Cancer Genome Atlas (TCGA): http://cancergenome.nih.gov; cBioPortal: http://www.cbioportal.org/index.do; Broad Institute Cancer Cell Line Encyclopedia (CCLE) https://portals.broadinstitute.org/ccle; Protein Data Bank (PDB): http://www.rcsb.org/pdb/home/home.do; Clustal Omega: (http://www.ebi.ac.uk/Tools/msa/clustalo/); SpliceAid2 (http://193.206.120.249/splicing_tissue.html); SFmap (http://sfmap.technion.ac.il/). Human Splicing Finder (HSF), www.umd.be/HSF3/; The R project for statistical computing: http://www.r-project.org/; Integrative Genomics Viewer (IGV): http://www.broadinstitute.org/igv; ASpli R package: https://bioconductor.org/packages/release/bioc/vignettes/ASpli/inst/doc/ASpli.pdf

**Reporting summary**. Further information on research design is available in the Nature Research Reporting Summary linked to this article.

## Data availability
The authors declare that data supporting the findings of this study are available from TCGA or within the paper and its supplementary information files. Source data for the main and supplementary figures are available in Supplementary Data 1–8. Unprocessed blot and gel images are provided in Supplementary Fig. 11. The RNA-seq dataset for T47D cells infected with SeV and uninfected cells has been deposited to NCBI Short Read Archive (SRA), accession number PRJNA512015. Additional information, protocols, and reagents can be provided on request to the corresponding author (L.P.-O.).

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

## Acknowledgements

The results presented here are in part based upon data generated by the TCGA Research Network. The study was supported by the Intramural Research Program of the National Cancer Institute - Division of Cancer Epidemiology and Genetics (L.P.-O.), AIDS Targeted Antiviral Program, Center for Cancer Research (V.K.P.), and the NCI Innovation Award (R.B.). We thank the Cancer Genomics Research Laboratory of NCI for RNA-sequencing, Nathan Cole for help with retrieval of sliced RNA-seq BAM files from the NCI Data Commons portal, and Dr. Marc-Henri Stern, Institute Curie, Paris, for critical comments and suggestions.

## Author contributions

A.R.B. and L.P.-O. conceived and designed the study. L.P.-O. supervised the study. A.R.B., O.O.O., S.H.-Y.L., A.O., J.M.V., K.A.D.-F., A.B., and C.Z. performed the experiments and analyzed the data. A.R.B., P.L., and O.F.-V. analyzed the RNA-seq data and performed statistical analyses. A.R.B., O.O.O., V.K.P., L.D., and L.P.-O. contributed reagents/materials/analysis tools. A.R.B. and L.P.-O. wrote the manuscript. All authors discussed the results and commented on the manuscript.

## Funding

## Competing interests

The authors declare no competing interests.
