## [Peer Review File · Communications Biology]

Reviewers' comments:

Reviewer #1 (Remarks to the Author):

The manuscript presents evidence for the role of alternative splicing in the regulation of mutagenic effects of APOBEC3A and 3B. The observations made the authors add new information about the regulation of editing deaminases. Also, the authors present valuable information relevant to the role of APOBEC3s in various cancers. High expression of APOBEC3s is associated with cancer. Exon 5 skipping in A3B results in an inactive enzyme. Inhibitors of the splicing factor can be used for the regulation of the mutagenicity of APOBECs.

The authors unambiguously analyzed RNA seq data concerning isoforms by identifying exon-exon reads. Alternative isoforms of A3A were detected in a small number of samples, but isoforms of A3B were seen in more than 50% of samples.

At this point, it would be good to have some rationale for the basis of the selection of the number of reads in groups and also the distribution of samples with one vs. ten reads in the group 2. Should the gates be different for the transcripts with or without frameshifting events?

Isoform-specific analysis refined the correlation of A3A and B expression with different tumor types. Computational and biochemical analysis revealed that isoforms that result from alternative splicing miss parts of the proteins necessary for deamination, and hence they are unable to catalyze the reaction in vitro.

A3B appeared to play a more distinct role in bladder cancer than A3A, and thus most of the paper describes its behavior. A3B levels positively correlated with increased mutation loads and shorter progression-free survival in patients with bladder cancer. Interesting, the proportion of alternatively spliced variants was lower in some tumors in comparison to normal tissues, suggesting that in normal tissues, the alternative splicing happens more frequently. The authors proposed and proved that the shift towards alternative splicing could be regulated by levels of splicing factors and facilitated by weak branch point sites in intron 4. Logically, the inhibitor of splice complex SF3b, pladienolide B, provoked alternative splicing, followed by the decay of nonsense codon containing DNA, thus reducing levels of a mutagenic form of A3B. The authors suggest that this could be used in "clinically relevant situations" but did not elaborate on the conditions that will benefit from such inhibition.

Comments

Lines 68-70. Reference to helpful Note S1 looks misplaced in the context of the sentence.

The authors should better explain the connection of the appearance of non-mutagenic forms of A3B2 or 3 and decay of nonsense codon containing mRNA. The level of the A3B2 and 3 detected in tumors are similar (Fig 1 C), but the A3B2 transcript is not supposed to be a substrate for NMD.

Line 166. ... "some factors" looks weak, undecisive for the section title.

Can the authors explain the rationale for using various doses (Fig 2C, 5-fold range: Fig. 6: 100-fold range) of inhibitors and comment on the variable or absent dose-effect relationships?

Fig.6 E is not easy to evaluate. We see the variable shape of curves, small symbols, and lack of the author's interpretations in the text. We also note that something is wrong with the legend to Fig 6 panel E. Why it starts with RT-PCR results, did the authors refer to qRT in panel D? The statement that "...splicing shift... resulting in downregulation of A3B1 protein, while the aberrant and unstable A3B3 protein was not detected" is confusing. Recombinant A3B3 looked OK in Fig. 2F (despite the undescribed but labeled by asterisk upper band) and in Notes S3 and S2, but it runs faster than A3B1 and will not be seen on the cut portion of the gel. Also, it looks that available antibodies to A3B do not recognize this form (Note S2). It is not clear what antibodies were used for this particular WB.

General impression about some western blots in the supplement of the manuscript is not very favorable, loading marker presence is highly variable, and thus some conclusions could not be firm.

Reviewer #2 (Remarks to the Author):

Brief summary of the manuscript

This work from Banday et al, provides an interesting mechanism for treatment of APOBEC-mediated mutagenesis through inducing exon skipping which results in a non-mutagenic isoform of APOBEC3A and 3B. The authors show that this non-mutagenic isoform is subjected to NMD and depletes the mutagenic isoform. The authors use an extensive number of methods including a clever "exon-trap" vector to show particular conditions which alter the splicing around exon 5 and 6. Interestingly, the altered splicing was more common in normal tissues than tumours, suggesting that this mechanism is an intrinsic mechanism to regulate APOBEC3B levels. Using the proposed mechanisms to "switch" the splicing to the non-mutagenic form could prove useful for developing therapies to treat cancers such as bladder cancer which have high levels of mutagenic APOBEC activity.

Overall impression of the work

This work represents an extensive profiling of alternate splicing of APOBEC3A and 3B isoforms. The authors present novel ideas and thoroughly investigate their hypotheses. Many of the comments below represent minor required changes, however there are a number of major improvements to be made before publication.

Major points:

1. You postulate that the sensitivity of A3B exon skipping to expression levels of different splicing factors (Line 190)- can you confirm this using RNAseq (if publicly available) for some cell lines, or using qPCR for the splicing factors? This would be nice to support your results. It would be useful to establish which SF is responsible in these cell lines, at least for the SFs you focus on, SF3B1 and SRSF2.

2. You specifically single out SF3B1 (Line 227) but do not include this in the panel of splicing factors assayed using the exontrap system. Is this because it is a subunit of the spliceosome? Can you overexpress SF3B1 and determine if over expression alters exon skipping?

3. A figure showing the impressive blocking of Bleomycin induction with pladienolide B with measurable levels of GAPDH and A3B is lacking. Fig 6G deamination assay results in too low concentration of protein. Repeating 6E with the same conditions but with bleomycin present would strengthen the result.

4. Referencing figures- label each panel with a letter, saying "left panel" or "top panel" is confusing when you have multi-panelled figures.

5. Figure legends need to be clearer and explain all acronyms used in the figure. Include n and what the error bars represent (SD vs SEM) in every figure with them.

6. There is a lot of supplementary data that is not clearly justified in the text. Include a little more detail in the main text when referencing the supplementary data (especially the "notes")

7. Figure 4. This figure needs improvements for clarity.

A Can you have a schematic above the first plots to show what is being measured? These plots are very large compared to the rest of the figures so could be resized. Insert (N) and (T) after normal and tumour in the legend. Clarify in the text as you say the alternative splicing is lower but the plots show higher values (because less is spliced sure, but could be worded better!)

B instead of an asterisk can you write N or T above each sample (also in C). This will clear it up somewhat. I think this is an important result that is being lost with unclear figures. I would also like arrows for what each band is where you have labelled only A3B1.

Specific minor points about figures:

- Fig 1. What is aE? You also state that "A3A1 is not expressed in most TCGA samples" when it is expressed in 63%. You don't mention A3B1. Make sure acronyms are correct "TGCA"
- Fig 2 A. A3AB appears here without any reference in the text or legend.

C Keep axis the same in comparable graphs. Why does expression of A3B3 decrease with increasing digoxin concentration?

F ensure the arrows align to the bars. Explain the asterisk

G not sure what this adds? It is not explained well and confusingly labelled. What is the luciferase expressed from?

8. Fig 3. What does the line and the shading represent? How was the correlation performed?

D I presume the grouping in each graph was performed for either A3B1 or A3A1, not just A3B1?

9. Fig 4. C Can this be repeated? The quality is quite poor.

D State that A3B2 was not quantifiable in the legend. What does "not quantifiable mean" not detectable? Not consistent?

10. Fig 5. I really like this experiment! Explain VE and E in the legend

A Why does CELF1-T4 appear to have a larger skipped band?

B Can you label what the cell lines are on the blot like you have for the plot (Bladder, breast etc) Start the Y axis at zero. Are any of these significant?

11. Fig 6. Typo in fig A (binding)

C very impressive induction of skipping of exon 5 and not 6!

D Can you hypothesise why the expression changes at 1 nM concentration are different (A3B1 decreases before returning to baseline and A3B3 goes drastically up then levels out/decreases).

G state in the legend that Bleomycin should induce A3B1 expression. Can you quantify this blot as well, cp to GAPDH, especially considering the low A3B1 expression?

12. Fig 7. The concept of SSOs seems a bit out of the blue given you don't show any data about this

Minor points in the text

Line 48 space before RNA-seq

Line 64 Mention A3AB here as it is in fig 2. Include A3AB schematic also in Fig S1A

Line 66 Include reference to TCGA sample table S2

Line 70- can you say something about the contents of "Note S1" in the text. It was a surprise to see the methods showing this RNAseq database that hadn't been mentioned at all in the text

Line 224 references Fig 1C when it should be 2C

Line 360 You include MCF7 (not used) but not Hela (S8) in the cell line list

Line 144 clarity here as you state the alternative splicing is lower because the canonical transcript is increased- a bit confusing when looking at the graph

Line 147, 151 Check all acronyms throughout, e.g. LIHC not LICH also in Fig 4A

Line 157 The differences between the normal and tumours are striking. You should write about the frequencies in each type here. E.g Alternative splicing was seen in xx% of normal and xx% of tumour.

Supplementary

Update Figure S1 to include A3AB

Line 349 anti tubulin is missing from the list (used in Fig 2G HIV assay)

December 14, 2020

We thank the reviewers for their thoughtful and constructive comments. We have updated the manuscript as summarized below and have provided point-by-point responses to the reviewers. The tracked version shows major changes are highlighted in yellow.

In the current version, we have:

1. Added data showing a correlation between *A3B* exon 5 splicing based on mini-gene assays and CCLE RNA-seq data for mRNA expression of eight splicing factors, for which we predicted binding sites within exon 5 (**Note S5**). The analysis was done in nine cell lines that we used for showing differential *A3B* exon 5 skipping, as presented in **Figure 5C**. This correlation analysis further confirmed the role of SRSF2 in causing skipping of *A3B* exon 5.
2. Added an experiment, shown as a Western blot (**Figure 6H**), demonstrating that pladienolide B decreases the bleomycin-induced *A3B* expression. This experiment supports the utility of pladienolide B to reduce *A3B* expression both at baseline (**Figure 6E**) and also treatment-induced, such as by bleomycin (**Figure 6G & H**).
3. Included new data on *A3B* exon 5 skipping caused by splice-switching oligos (SSOs) (**Figure 7A & B**). We tested SSOs for *A3B* branch points sites within intron 4 and for 3' splicing site of exon 5. Targeting the splicing site by SSO caused up to 80% of exon 5 skipping. These results justify further efforts for using SSOs to decrease the expression level of the mutagenic *A3B* isoform.
4. Modified text and figures in multiple places, as suggested by reviewers.
5. Modified the title to best highlight the main message of the work, which now reads as "Isoform-specific characterization implicates alternative splicing in *APOBEC3B* as a targetable mechanism for restricting APOBEC-mediated mutagenesis".

Point-by-point responses:

Reviewer #1 (Remarks to the Author):

The manuscript presents evidence for the role of alternative splicing in the regulation of mutagenic effects of APOBEC3A and 3B. The observations made the authors add new information about the regulation of editing deaminases. Also, the authors present valuable information relevant to the role of APOBEC3s in various cancers. High expression of APOBEC3s is associated with cancer. Exon 5 skipping in A3B results in an inactive enzyme. Inhibitors of the splicing factor can be used for the regulation of the mutagenicity of APOBECs.

The authors unambiguously analyzed RNA seq data concerning isoforms by identifying exon-exon reads. Alternative isoforms of A3A were detected in a small number of samples, but isoforms of A3B were seen in more than 50% of samples.

At this point, it would be good to have some rationale for the basis of the selection of the number of reads in groups and also the distribution of samples with one vs. ten reads in the group 2. Should the gates be different for the transcripts with or without frameshifting events?

Isoform-specific analysis refined the correlation of A3A and B expression with different tumor types. Computational and biochemical analysis revealed that isoforms that result from alternative splicing miss parts of the proteins necessary for deamination, and hence they are unable to catalyze the reaction in vitro.

A3B appeared to play a more distinct role in bladder cancer than A3A, and thus most of the paper describes its behavior. A3B levels positively correlated with increased mutation loads and shorter progression-free survival in patients with bladder cancer. Interesting, the proportion of alternatively spliced variants was lower in some tumors in comparison to normal tissues, suggesting that in normal tissues, the alternative splicing happens more frequently. The authors proposed and proved that the shift towards alternative splicing could be regulated by levels of splicing factors and facilitated by weak branch point sites in intron 4. Logically, the inhibitor of splice complex SF3b, pladienolide B, provoked alternative splicing, followed by the decay of nonsense codon containing DNA, thus reducing levels of a mutagenic form of A3B. The authors suggest that this could be used in “clinically relevant situations” but did not elaborate on the conditions that will benefit from such inhibition.

Response: We thank the reviewer for the supportive comments. There are two general concerns:

“At this point, it would be good to have some rationale for the basis of the selection of the number of reads in groups and also the distribution of samples with one vs. ten reads in the group 2. Should the gates be different for the transcripts with or without frameshifting events?”

Response: The distribution of samples in relation to the number of reads is arbitrary (**Fig. 1**). We present it just for the purpose of data visualization. Because of its arbitrary nature, we do not present statistical analyses between the transcript-specific sample groups. Also, in this analysis, we do not infer the functionality of the transcript or protein isoforms. Of all the transcript isoforms, only *A3B3* has a frameshift, which is already presented separately in **Fig. 1** and **Fig. S1B**.

The authors proposed and proved that the shift towards alternative splicing could be regulated by levels of splicing factors and facilitated by weak branch point sites in intron 4. Logically, the inhibitor of splice complex SF3b, pladienolide B, provoked alternative splicing, followed by the decay of nonsense codon containing DNA, thus reducing levels of a mutagenic form of A3B. The authors suggest that this could be used in “clinically relevant situations” but did not elaborate on the conditions that will benefit from such inhibition.

Response: We mentioned possible clinical implications in several places, but we discuss this in detail in the Discussion:

“Therapeutic targeting of AS to treat cancer is a rapidly developing field⁵⁰ with several types of SF3B1 inhibitors, such as E710751,⁵² an analog of pladienolide B, and H3B-880053 being evaluated for acute myeloid leukemia (AML) and myelodysplastic syndrome (MDS)^{33,50,53}. Because tumor cells depend on wild-type SFs for survival, these drugs are particularly effective in killing cancer cells with already impaired splicing machinery due to mutations in SFs, such as SRSF2 and SF3B1⁵³. Mutations in these and other SFs have been reported in many solid tumors, including bladder cancer⁵⁴ and these tumors tend to have more APOBEC-signature mutations (Figure S10). Our results suggest that SF3B1 inhibitors and other tools affecting alternative splicing might be tested to eliminate cancer cells with mutations in SFs and also to control APOBEC mutagenesis in clinically relevant conditions. It is interesting to note that FGFR3-S249C, the most common somatic mutation found in the highly recurrent non-muscle-invasive bladder cancer, is likely caused by APOBEC mutagenic activity⁵⁵. As we found A3B1 the primary mutagenic APOBEC enzyme in bladder tumors, enhancing A3B exon 5 skipping might help to restrict APOBEC-mediated mutagenesis, including FGFR3-S249C mutation in bladder cancer, as well as prevent tumor progression and recurrence, clonal evolution, and resistance to chemotherapy (Figure 8). In addition to using SF3B1 inhibitors, we also demonstrate the utility of SSOs for inducing exon 5 skipping and decreasing the expression of mutagenic A3B isoform, possibly opening up avenues for targeting APOBEC-mediated mutagenesis by oligo-based drugs⁵⁶. Notably, targeting any other gene region of the highly homologous APOBEC3 gene family (Note S1) might cause off-target effects, while using SSOs uniquely increasing the skipping of exon 5 of A3B would result in natural NMD-based elimination of the non-functional A3B3 isoform produced in this process.”

Comments

Lines 68-70. Reference to helpful Note S1 looks misplaced in the context of the sentence. **Response:**

To improve clarity, we now replaced this sentence to mention **Note S1** only in one place:

“In Note S1, we show that due to high homology between A3A and A3B, quantification based on specific exon-exon junctions (by RNA-seq or qRT-PCR) is more reliable than based on total gene expression analysis (by RNA-seq). The misaligned RNA-seq reads would incorrectly represent the expression of both genes, affecting downstream analyses and biological interpretations”.

The authors should better explain the connection of the appearance of non-mutagenic forms of A3B2 or 3 and decay of nonsense codon containing mRNA. The level of the A3B2 and 3 detected in tumors are similar (Fig 1 C), but the A3B2 transcript is not supposed to be a substrate for NMD.

Response: In **Fig. S1B**, we present exon reads for both canonical and alternative splicing events from whole TCGA data and show read distribution in **Fig. 1C**. Although splicing reads for A3B3 (E4-56) appear more abundant compared to A3B2 (E5-aE6), the overall trend is similar for both. We do not have a good explanation as to why the levels of the two isoforms appear similar in TCGA. However, our qRT-PCR and RT-PCR data in a separate dataset of bladder tumors (see **Fig. 4B**) suggested that A3B2 is rarely expressed compared to A3B3, even though the latter is a potential target for NMD. In all, alternative isoforms are expressed at very low levels (mean read count = 5, in samples that show expression) compared to A3B1 (mean read count = 49, in samples that show expression). This suggests that expression levels of alternative isoforms might not cause any direct phenotypic effects. But instead, the role of these alternative isoforms might be to reduce the expression of the mutagenic A3B1, suggesting a strategy of how A3B1 levels could be reduced, which we explored experimentally.

Line 166. ... "some factors" looks weak, undecided for the section title.

Response: Thanks for pointing this out. We have changed the section title as “*A3B exon 5 skipping is sensitive to expression levels of splicing factors.*”

Can the authors explain the rationale for using various doses (Fig 2C, 5-fold range: Fig. 6: 100-fold range) of inhibitors and comment on the variable or absent dose-effect relationships?

Response: We used these concentrations based on previous publications - for digoxin (PMID:23711097; PMID: 25193633) and pladienolide B (PMID: 24635824; PMID: 29796170). For digoxin treatment shown in **Fig. 2C**, we observed increased expression levels for *A3B3* with increased digoxin concentrations. The goal of this experiment was to demonstrate the effect of inhibition of nonsense-mediated decay pathway on *A3B3* expression. Treatments with all three concentrations of digoxin resulted in higher levels of *A3B3* compared to DMSO control, proving that *A3B3* expression is indeed affected by NMD.

We observed a dose-dependent effect on *A3B1* levels at lower concentrations of pladienolide B (1, 10, 20 nM). However, the effect was similar between 20 and 40 nM concentrations and slightly decreased at 100 nM. Compared to *A3B1*, the effect on *A3B3* levels at different concentrations was not consistent, probably because *A3B3* is eliminated by NMD. It appears that *A3B* splicing changes become unresponsive to pladienolide B concentrations after a certain threshold. It could be because general splicing disruption at higher concentrations could cause increased cell death. Previous studies also observed similar results for the effect of pladienolide B on splicing modules of other genes (PMID: 24635824).

Fig.6 E is not easy to evaluate. We see the variable shape of curves, small symbols, and lack of the author's interpretations in the text.

Response: We agree that the shapes of curves for *A3B3* expression with error bars in **Fig. 6D** look inconsistent. We think that error bars are wide because the rate of *A3B3* degradation by NMD might be sensitive to pladienolide B indirectly, resulting in higher variability at low expression levels of *A3B3*. We have tried to improve these qRT-PCR results but were not successful. However, the consistent reduction in *A3B1* expression (both RNA and protein, **Fig. 6E**) is more important than the increase in *A3B3* expression. We have modified the text as:

“Skipping of exon 5 in endogenous A3B was also increased by pladienolide B treatment, resulting in increased expression of A3B3 mRNA and decreased expression of A3B1 mRNA and A3B1 protein (Figure 6D, E, Figure S8A). Importantly, the reduction of A3B1 protein caused by pladienolide B treatment was more prominent and consistent than the induction of A3B3 expression. A3B3 expression was much higher in cells treated with pladienolide B in the presence of digoxin, an NMD blocker21 (Figure 6F), supporting NMD-based degradation of A3B3 transcript (Figure 1C) as a mechanism contributing to its poor detection on mRNA level and no detection on the protein level.”

We also note that something is wrong with the legend to Fig 6 panel E. Why it starts with RT-PCR results, did the authors refer to qRT in panel D?

Response: We thank the reviewer for pointing out the mistake in the legend, which we now revised as: *“Results of qRT-PCR in HT-1376 cells treated with increasing concentrations of pladienolide B for indicated time points show splicing shift from A3B1 to A3B3.”*

The statement that “...splicing shift... resulting in downregulation of A3B1 protein, while the aberrant and unstable A3B3 protein was not detected” is confusing. Recombinant A3B3 looked OK in Fig. 2F (despite the undescribed but labeled by asterisk upper band) and in Notes S3 and S2, but it runs faster than A3B1 and will not be seen on the cut portion of the gel. Also, it looks that available antibodies to A3B do not recognize this form (Note S2). It is not clear what antibodies were used for this particular WB.

Response: We appreciate the thoughtful comments of the reviewer. We have corrected the statement as “Western blot protein analysis of HT-1376 cells treated with increasing concentrations of pladienolide B for 36 hrs showed downregulation of A3B1 protein, while expression of endogenous A3B3 protein was not detected.”

We tested multiple commercial antibodies for detecting the recombinant A3B3 protein by Western blot but were not successful. In **Fig. 2F** and Notes **S2** and **S3**, recombinant A3B3 is detected by an anti-DDK/Flag antibody for the tag. We could not detect the endogenous protein, which is expressed at lower levels and is not recognized by any available antibodies for A3B1.

We could not confirm the identity of the upper band marked by an asterisk in the lanes with A3B3 recombinant protein (**Fig. 2F**). Because this band did not appear in Western blots presented for detecting partially purified proteins (**Note S3**), we think it could be a nucleotide probe bound with A3B3 in the deamination reaction mix. Western blot presented in **Fig. 2F** is based on deamination reactions.

General impression about some western blots in the supplement of the manuscript is not very favorable, loading marker presence is highly variable, and thus some conclusions could not be firm.

Response: We assume this comment is about Western blots presented in **Note S2** and **S3**. These gels are for antibody testing and identification of partially purified recombinant proteins. The size differences are due to the use of different loading markers during the purification process. We are certain about the identity of the recombinant proteins because these proteins were detected by an anti-DDK/Flag antibody as presented in **Fig 2.** and in **Figs. S2 & S3.**

Reviewer #2 (Remarks to the Author):

Brief summary of the manuscript

This work from Banday et al, provides an interesting mechanism for treatment of APOBEC-mediated mutagenesis through inducing exon skipping which results in a non-mutagenic isoform of APOBEC3A and 3B. The authors show that this non-mutagenic isoform is subjected to NMD and depletes the mutagenic isoform. The authors use an extensive number of methods including a clever “exon-trap” vector to show particular conditions which alter the splicing around exon 5 and 6. Interestingly, the altered splicing was more common in normal tissues than tumours, suggesting that this mechanism is an intrinsic mechanism to regulate APOBEC3B levels. Using the proposed mechanisms to “switch” the splicing to the non-mutagenic form could prove useful for developing therapies to treat cancers such as bladder cancer which have high levels of mutagenic APOBEC activity.

Overall impression of the work

This work represents and extensive profiling of alternate splicing of APOBEC3A and 3B isoforms. The authors present novel ideas and thoroughly investigate their hypotheses. Many of the comments below represent minor required changes, however there are a number of major improvements to be made before publication.

Response: We thank the reviewer for the constructive comments and the appreciation of the significance of our work.

Major points:

1. You postulate that the sensitivity of A3B exon skipping to expression levels of different splicing factors (Line 190)- can you confirm this using RNAseq (if publicly available) for some cell lines, or using qPCR for the splicing factors? This would be nice to support your results. It would be useful to establish which SF is responsible in these cell lines, at least for the SFs you focus on, SF3B1 and SRSF2.

Response: Thanks for this suggestion. We now performed this analysis and the results are discussed as:

“ Screening of exon 5 splicing in 10 cell lines of different tissue origin also showed variable exon 5 skipping (Figure 5C), presumably due to differences in levels of expression/activity of endogenous SFs in these cell lines. To test this, we performed a correlation analysis between the percentage of A3B exon 5 skipping in the mini-gene experiment (Figure 5C) and total mRNA expression of the eight SFs evaluated in this experiment (Figure 5C), using RNA-seq data available from the Cancer Cell Line Encyclopedia (CCLE). HEK293T cell line was excluded from this analysis because RNA-seq data for this cell line was not available in CCLE. A3B exon 5 skipping showed significant positive correlations with expression of SRSF2, KHDRBS1 and MBNL1, and negative correlations with expression of ELAVL2 (Note S5). The correlation with SFRS2 expression was the strongest and supported the results of the overexpression model (Figure 5B). Thus, our results suggested that A3B exon 5 skipping is sensitive to expression levels of multiple SFs, which can bind to cis-regulatory motifs within this exon.”

2. You specifically single out SF3B1 (Line 227) but do not include this in the panel of splicing factors assayed using the exontrap system. Is this because it is a subunit of the spliceosome? Can you overexpress SF3B1 and determine if over expression alters exon skipping?

Response: For Exontrap analysis, we selected only splicing factors with binding sites predicted within exon 5 (Table S3). SF3B1 is a subunit of the spliceosome, which interacts with intronic branch point sites (BPS). We suspected that SF3B1 might be involved when Exontrap/mini-gene for two inserts (with and without predicted BPS) of exon 5 indicated potential weak BPS in intron 4 (Fig. 5C). Given that the cell lines express mainly A3B1 isoform, overexpression of SF3B1 would not be as appropriate a model as using the knockdown. In fact, we confirmed the role of SF3B1 in exon 5 splicing by siRNA knockdown, which is shown in Fig. S8B. The sequestering of SF3B1 with pladienolide B or by siRNA knockdown shifts splicing towards A3B3, thus proving the role of the BPS in facilitating exon 5 skipping.

3. A figure showing the impressive blocking of Bleomycin induction with pladienolide B with measurable levels of GAPDH and A3B is lacking. Fig 6G deamination assay results in too low concentration of protein. Repeating 6E with the same conditions but with bleomycin present would strengthen the result.

Response: Thanks for the comment. We have included the Western blot (Fig. 6H) that shows that pladienolide B decreases the bleomycin-induced A3B expression levels.

Pladienolide B is a generic splicing inhibitor and specific experiments will be needed to validate its utility for controlling APOBEC-mediated mutagenesis. As an alternative approach, also utilizing the knowledge on A3B exon 5 splicing, we performed an additional experiment using splicing-switching oligos (SSOs) for BPS and 3' splicing site of exon 5 to cause skipping of exon 5 as a direct way of reducing mutagenic A3B1 protein. Based on these results, we report an SSO-3'SS caused up to 80% of exon 5 skipping (Fig. 7A, B).

4. Referencing figures- label each panel with a letter, saying “left panel” or “top panel” is confusing when you have multi-panelled figures.

Response: Thank you for this suggestion. We have now labeled each panel and removed the left/top panel designations.

5. Figure legends need to be clearer and explain all acronyms used in the figure. Include n and what the error bars represent (SD vs SEM) in every figure with them.

Response: We have modified legends, added ‘n’ and explained error bars wherever they were missing or unclear.

6. There is a lot of supplementary data that is not clearly justified in the text. Include a little more detail in the main text when referencing the supplementary data (especially the “notes”)

Response: Thanks for this suggestion. We have improved the explanation of the supplementary materials, including Notes in the main text.

7. Figure 4. This figure needs improvements for clarity.

A Can you have a schematic above the first plots to show what is being measured? These plots are very large compared to the rest of the figures so could be resized. Insert (N) and (T) after normal and tumour in the legend. Clarify in the text as you say the alternative splicing is lower but the plots show higher values (because less is spliced sure, but could be worded better!)

B instead of an asterisk can you write N or T above each sample (also in C). This will clear it up somewhat. I think this is an important result that is being lost with unclear figures. I would also like arrows for what each band is where you have labelled only A3B1.

Response: We implemented the suggested changes to improve **Fig. 4** and its legend. Specifically:

- Added the schematics to panels A and B
- Resized the plots
- Inserted (N) and (T) corresponding to “normal” and “tumor” samples in the legend
- Changed the text for “alternative splicing is lower”
- Replaced asterisk with T and labeled normal lanes with N.
- Labeled bands for each isoform

Specific minor points about figures:

- Fig 1. What is aE? You also state that “A3A1 is not expressed in most TCGA samples” when it is expressed in 63%. You don’t mention A3B1. Make sure acronyms are correct “TGCA”

Response: aE refers to an alternative exon. The **Fig. 1** legend is now updated as:

“‘E’ refers to exon and ‘aE’ refers to an alternative exon. Based on exon junction reads (≥ 1 read/sample), A3A1 (E1-E2 junction) and A3A2 (E1-aE2 junction) is expressed in 6979 (63.10%) and 386 (3.49%) TCGA samples, respectively. C) Schematics of A3B exons and splicing junctions and comparison of gene expression based on the total and exon-exon junction reads corresponding to A3B isoforms. A3B1 is expressed in 10572 (95.8%) TCGA samples. But A3B2 and A3B3 are expressed in 53.55% (5,922 of 11,058) and 51.84% (5,733 of 11,058) of TCGA samples, respectively. The TCGA RNA-seq set includes 10,328 tumors and 730 adjacent normal tissue samples.

- Fig 2A. A3AB appears here without any reference in the text or legend.

Response: Thanks for pointing this out. We have removed A3AB as it is not a splice variant but created by the germline deletion.

Fig 2C Keep axis the same in comparable graphs. Why does expression of A3B3 decrease with increasing digoxin concentration?

Response: All comparable graphs of **Fig. 2C** are updated with the same scale for X-axis. We think the decrease in A3B3 expression is probably due to the secondary effects of digoxin at higher concentrations, which might be resulting in cellular toxicity.

Fig 2F ensure the arrows align to the bars. Explain the asterisk

Response: We updated the panel **2F**. Asterisk is now explained in the legend as:

“Asterisk indicates a non-specific band resulting from deamination of ssDNA probe by A3B3. Because DNA binding domain is intact in A3B3, the higher MW band could indicate a probe bound by A3B3.”

G not sure what this adds? It is not explained well and confusingly labelled. What is the luciferase expressed from?

Response: Panel **2G** supports the results for HIV infectivity assay for isoforms presented in **Fig. S3**.

Luciferase is expressed from TZM-bl cells containing HIV-1 Tat-inducible luciferase reporter gene. It is described in the methods section as:

“To generate the virus for infection, HEK293T cells (4×10^5 cells/ 6-well dish) were transfected using LT1 reagent (Mirus Bio) with HDV-EGFP (1 ug), pHCMV-G (0.25 ug), and variable concentrations of plasmids (0, 680 and 1200 ng), individually or in combinations. The virus was harvested 48 hrs post-infection, filtered with 0.45-um-pore filters, and stored at -80°C . Capsid p24 measurements were determined using the HIV-1 p24 capsid (CA) ELISA Kit (XpressBio). Normalized p24 CA amounts were used to infect TZM-bl cells containing HIV-1 Tat-inducible luciferase reporter gene, in a 96-well plate (4000 cells/well). Luciferase activity was measured 48 hrs after infection, using a 96-well luminometer (LUMIstar Galaxy, BMG LABTECH). For some experiments, portions of the viral supernatant were spun through a 20% sucrose cushion (15,000 rpm, 2 hrs, 4°C , in a Sorvall WX80 + ultracentrifuge), concentrated 10-fold, and used in experiments to determine virion encapsidation of APOBEC3 proteins by Western blotting analysis as previously described²³.”

8. Fig 3. What does the line and the shading represent?

Response: The line or “best-fit line” represents the linear trend in the scatterplots for correlation and the shading represents the 95% confidence interval around the line of best fit. We have included this information in the legend.

How was the correlation performed?

Response: Correlation analysis was performed with the Spearman method. We have included this information in the legend.

D I presume the grouping in each graph was performed for either A3B1 or A3A1, not just A3B1?

Response: Yes, the grouping was performed for both A3A1 and A3B1.

9. Fig 4. C Can this be repeated? The quality is quite poor.

Response: Unfortunately, we are unable to repeat this experiment because of the lack of protein extracts from these primary human tumors.

D State that A3B2 was not quantifiable in the legend. What does “not quantifiable mean” not detectable? Not consistent?

Response: We have updated the legend. As shown by PCR in **Fig. 3B** (now as **3C**), A3B2 is rarely expressed in these bladder tumors and adjacent normal tissues and thus qRT-PCR data on the expression of A3B2 in few samples was not comparable with A3B1. This is what we refer to as ‘not quantifiable’.

10. Fig 5. I really like this experiment! Explain VE and E in the legend

Response: We thank the reviewer. V and E stand for vector and exon, respectively. We have now explained these acronyms in the legend.

A Why does CELF1-T4 appear to have a larger skipped band?

Response: CELF1-T4 overexpression might have activated a new cryptic splice site. However, we did not explore this observation outside of our main research question.

B Can you label what the cell lines are on the blot like you have for the plot (Bladder, breast etc) Start the Y axis at zero.

Response: We have updated the **Fig. 5B** (now **Fig. 5C**) as suggested.

Are any of these significant?

Response: In **Fig. 5C**, the comparison of exon 5 inclusion or exclusion is within cell lines. Because the effects might be cell line-specific, we did not calculate the significance of differences between cell lines.

11. Fig 6. Typo in fig A (binding)

Response: Addressed

C very impressive induction of skipping of exon 5 and not 6!

D Can you hypothesise why the expression changes at 1 nM concentration are different (A3B1 decreases before returning to baseline and A3B3 goes drastically up then levels out/decreases).

Response: SF3B1 is a core splicing factor and concentration-based inhibition of its function by pladienolide B appears very dynamic. The detailed analysis of complex splicing mechanisms regulated by SF3B1 is outside the scope of our paper. Exploring these mechanisms in relation to regulation of A3B splicing, we think at the lower concentrations, pladienolide B may be inducing another splicing event as can be seen by a band above A3B1 (lane with 10 uM pladienolide B, **Fig. 6C**), and this event consumes pre-mRNA leading to reduced expression of A3B1. For A3B3, it appears that maximum splicing is achieved at pladienolide B concentration between 20-40 uM, possibly consuming all A3B pre-mRNA and

not causing any more effects at 100 uM. It appears that pladienolide B can inhibit SF3B1 only to a certain extent because the severe loss of SF3B1 can be toxic to cells.

G state in the legend that Bleomycin should induce A3B1 expression. Can you quantify this blot as well, cp to GAPDH, especially considering the low A3B1 expression?

Response: We have now included an additional experiment, represented by Western blot (**Fig. 6H**). All Western blots in Fig. 6 are visually clear where the band corresponding to A3B shows an upregulation or downregulation trend with treatment groups. Respectfully, we think that normalization by GAPDH is not essential here.

12. Fig 7. The concept of SSOs seems a bit out of the blue given you don't show any data about this

Response: We have now added an experiment showing the utility of SSOs for modulating A3B exon 5 skipping (**Fig. 7A**).

Minor points in the text
Line 48 space before RNA-seq

Response: Addressed

Line 64 Mention A3AB here as it is in fig 2. Include A3AB schematic also in Fig S1A

Response: We have removed it from Fig. 2 because *A3AB* is not a splice variant but a result of a germline variation

Line 66 Include reference to TCGA sample table S2

Response: Addressed

Line 70- can you say something about the contents of "Note S1" in the text. It was a surprise to see the methods showing this RNAseq database that hadn't been mentioned at all in the text

Response: Addressed. We have modified the text as:

"We show that due to high homology between A3A and A3B, quantification based on specific exon-exon junctions (by RNA-seq or qRT-PCR) is more reliable than based on total gene expression analysis (by RNA-seq, Note S1). The misaligned RNA-seq reads would incorrectly represent the expression of both genes, affecting downstream analyses and biological interpretation."

Line 224 references Fig 1C when it should be 2C

Response: Addressed

Line 360 You include MCF7 (not used) but not HeLa (S8) in the cell line list

Response: Thanks for pointing this out. This issue is now addressed.

Line 144 clarity here as you state the alternative splicing is lower because the canonical transcript is increased- a bit confusing when looking at the graph

Response: Addressed as:

“We observed that the proportion of A3B canonical splicing was significantly higher, while the proportion of alternative splicing was significantly lower in tumors compared to corresponding adjacent normal tissues. Specifically, the proportion of A3B2 splicing was lower in tumors of KICH ($P = 1.0E-03$) and LUSC ($P = 3.30E-02$) (Figure 4A) and the proportion of A3B3 splicing was lower in tumors of BLCA ($P = 4.60E-03$), HNSC ($P = 6.50E-03$), LIHC ($P = 1.90E-02$), LUAD ($P = 1.50E-03$), and LUSC ($P = 3.89E-05$) (Figure 4B). All other cancer types showed no significant differences in this analysis (Figure S5B, C).”

Line 147, 151 Check all acronyms throughout, e.g. LIHC not LICH also in Fig 4A

Response: Addressed

Line 157 The differences between the normal and tumours are striking. You should write about the frequencies in each type here. E.g Alternative splicing was seen in xx% of normal and xx% of tumour.

Response: Addressed

Supplementary

Update Figure S1 to include A3AB

Response: As A3AB is not a splice variant so we have removed it from the main **Fig. 2**. So, not necessary to include it in **Fig. S1**.

Line 349 anti tubulin is missing from the list (used in Fig 2G HIV assay)

Response: Addressed

REVIEWERS' COMMENTS:

Reviewer #1 (Remarks to the Author):

The revision answers the critique. The authors undertook a substantial effort to modify the work. The manuscript is an important contribution to the literature.

Reviewer #2 (Remarks to the Author):

Thank you for updating the manuscript. Most of the suggested changes were included and overall I am happy to see this published. However there still remain a number of inconsistencies that need to be addressed prior to publication.

The inclusion of the RNAseq from CCLE was nice support for SRSF2 and even nice to see ELAVL2 show a negative correlation as in the exontrap experiments.

The SSO addition although not required was also very clear.

As a general note I highly recommend better track changes next time, there were many things altered that were not highlighted making it a painful re-review.

Most of my concerns that were not addressed relate to the figures and figure legends.

Figure legends still need to include what the plots and error bars represent. Graphs are meaningless unless you specify what is plotted. Any time you plot error bars state what they represent. This needs to be looked at for Fig 2C 2G 5B 5C 6D. This should be looked at in the supplementary as well.

Fig 4 A B and E you state "red bars represent mean expression levels" when surely they are mean +/- SD (or SEM)

If the intention is to show protein expression for tumour and not normal then the 4D western blot should be labelled as 4C (x of 63 and x of normals x of tumours)

Also 6E legend states protein quantitation is from 3 experiments, but there is no protein quantitation its just a WB?

The additional western blot (7H) showing the blocking of bleomycin induction with pladienolide B was nice to see, however can you state in the legend how much protein was loaded here as you mention the low lysate level for 7G.

Also where you have schematics (Fig 1 and 4) could you indicate the alternate exon splice site in a different colour, in 4 particularly it is not visible and looks too similar to the normal exon.

There also remains incorrect acronyms LICH appears at least twice (find LICH replace LIHC)